# Prototype of a System for Tracking Transit Service Based on IoV, ITS, and Machine Learning

**Camilo Andrés Sánchez Díaz** [1], **Andersson Stive Díaz Lucio** [1], **Ricardo Salazar-Cabrera** [1,*],
**Álvaro Pachón de la Cruz** [2] **and Juan Manuel Madrid Molina** [2]

1   Telematics Engineering Research Group (GIT), Telematics Department, Universidad del Cauca,
    Popayán 190003, Colombia; sdcamilo@unicauca.edu.co (C.A.S.D.); anderssond@unicauca.edu.co (A.S.D.L.)
2   Information Technology and Telecommunications Research Group (I2T), ICT Department, Universidad Icesi,
    Cali 760001, Colombia; alvaro@icesi.edu.co (Á.P.d.l.C.); jmadrid@icesi.edu.co (J.M.M.M.)
*   Correspondence: ricardosalazarc@unicauca.edu.co

**Abstract:** The transit service in a city should be the most efficient, least polluting, most accessible, and sustainable means of transportation for its citizens. However, serious shortcomings have been detected, mainly in medium-sized cities in developing countries. These shortcomings are related to a lack of user information, insecurity, low service availability, and repeated stops in inappropriate and/or unauthorized places. Some of these shortcomings contribute to high accident rates and traffic congestion. The development of tools to improve the characteristics and conditions of transit service in cities has become an imperative need to improve the quality of life of citizens and city sustainability. Transit service tracking is relevant in aspects such as online location information to travelers and control by transport companies for compliance with speed limits, schedules, routes, and stops. This research proposes a transit vehicle tracking system based on the Internet of Vehicles (IoV) in Vehicle-to-Roadside (V2R) classification. The proposed system is ideal for the use of electric vehicles due to the low power consumption of the tracking device. This system uses Intelligent Transportation Systems (ITS) tracking service architecture, Long Range (LoRa) communication technology, and its LoRa Wide Area Network (LoRaWAN) protocol. Additionally, the system offers real-time location prediction in the absence of position data. The IoV tracking device integrates a GPS-LoRa module card with an Inertial Measurement Unit (IMU). A location prediction algorithm was implemented to train and store a prediction model with previously collected data from tracking devices. To evaluate the developed model, a case study in the city of Popayán (Colombia) was implemented, using three routes for testing. The results of the system implementation were satisfactory, obtaining an average coverage of 60.4% of the routes in the final field tests through LoRa communication. For the remaining 39.6% of the routes, location data prediction was used, with an average accuracy of 177 m with respect to the real location. Considering the obtained results, a tracking system such as the one proposed in this article can be used in the transit systems of medium-sized cities in developing countries to improve service quality and fleet control.

**Keywords:** transit service; transit vehicle tracking; location prediction; LoRaWAN; machine learning

## 1. Introduction

The adequate control and tracking of transit vehicles in a city can encourage the use of public transportation and, thus, contribute to the reduction of traffic. This is more significant and beneficial for medium-sized cities in developing countries, such as Colombia. In these cities, the main transit service (known as "collective") shares the road with other types of vehicles and often does not have adequate control [1].

To track and control the vehicles in the "collective" transit service, it is viable to use Information and Communication Technology (ICT), such as location technologies, communications technologies, Machine Learning (ML), and Intelligent Transportation Systems (ITS).

Regarding location technologies, the Global Positioning System (GPS) continues to be the most used technology, with good results; although, there are some problems with service availability in certain areas of developing countries. Bluetooth is a location technology that has been used in some research related to vehicle tracking; but, the need for considerable infrastructure in cities makes it not a suitable option.

Currently, the communication technologies used for tracking transit vehicles are cellular networks, such as the Global System for Mobile Communications (GSM) and Long Term Evolution (LTE); but, there are some drawbacks regarding costs and technical aspects, such as range and coverage [2], which can be solved with Low Power Wide Area Networks (LPWANs).

A LPWAN is a type of communication technology that can be used in transit services. LPWANs are characterized by low power consumption, long range, and the transmission of small data packets over long distances [3]. Among these technologies is Long Range (LoRa). LoRa is a radio frequency modulation technique derived from the Chirp Spread Spectrum (CSS), which allows it to reach great distances and serve as a support for sending and receiving information in IoV-type scenarios, mainly in electric transit vehicles due to their low power consumption. LoRa is commonly used with its Media Access Control (MAC) protocol, known as LoRa in Wide Area Networks (LoRaWANs). The use of LoRaWANs allows for improved transmission and reception when multiple devices are operating at the same time in a reduced area. A LoRaWAN serves as a network protocol that manages the communication between gateways and end devices [4,5]. Some examples of the use of these technologies (LoRa and LoRaWAN) in vehicle tracking systems can be seen in [2,6–8].

These communication technologies (LoRa and LoRaWAN) apply to all types of transit vehicles (electric, hybrid, or gasoline vehicles). However, in electric or hybrid vehicles, the use of LoRa is ideal due to the low power consumption of the IoV devices [9–11].

The principal advantages of LoRa are the long range (several km), low power consumption, large number of messages per gateway, and low operation and infrastructure [4]. A LoRaWAN offers advantages, such as high security, bidirectional communication, and different frequency bands according to the geographical area [5].

The disadvantages of LoRa and LoRaWANs include requiring a Line of Sight (LoS) between the gateway equipment and the terminal devices, low data transmission speed, and small packet size (255 bytes) [5]. The need for a constant LoS between gateways and terminal devices when using LoRa in vehicle tracking systems generates coverage problems. Thus, it is necessary to implement some type of solution to solve the problem of vehicle location when there is no LoS.

ML is an interesting ICT option, which could be applied to solve the problem of vehicle location when there is no LoS when using a communication technology such as LoRa. Location prediction using ML is an option applied in some vehicle tracking research [12–18]. ML is used in these works to predict the future location of vehicles through a collection of data, such as latitude, longitude, and stops.

Finally, the concept of ITS must be considered when evaluating the ideal technologies to apply in a vehicle tracking system. The use of ITS as a basis for the development of this type of system is ideal for achieving standardization and interoperability between the different types of mobility services that are developed for a particular city and taking advantage of the use of new technologies in aspects such as communication and tracking [19]. The most used ITS architectures worldwide are The American Architecture Reference for Cooperative and Intelligent Transportation (ARC-IT) [20] and the European Intelligent Transport Systems Framework Architecture [21], which propose the components, interactions, and communications necessary for implementing mobility services identified as relevant in different areas. The transit vehicle tracking service (in the public transportation area) is one of the services proposed by these architectures, presenting the specific architecture recommended to implement it properly [19].

Considering the above, the development of a prototype of a system for tracking service in real-time, based on IoV, ITS, and ML, was proposed, hoping this system can improve the

characteristics of the monitoring systems of the "collective" transit service in the medium-sized cities of developing countries (from now on, the target context); encourage its use by citizens; and, indirectly, contribute to the reduction of vehicular congestion and accidents.

The novelty of this work is identified in the following aspects. The proposed prototype allows the tracking and monitoring of transit vehicles and is focused on developing countries (where transit services are often not adequately controlled), which is a context with very little research in this respect. For this monitoring, the prototype uses a low-power IoV device to capture relevant vehicle data and communicate it via a LoRaWAN to a cloud server. A LoRaWAN is a communication technology not commonly used in this type of system; but, it is an option with good results in recent years and savings in operating costs when compared to conventional technologies, such as the GSM or LTE, that are used in most related works. The prototype also proposes the use of ML to predict vehicle location when there is no line of sight between gateways and terminal devices. The use of a reference ITS architecture for the transit vehicle tracking service is also a relevant aspect. Finally, the validation of the developed prototype was performed through field tests instead of simulation, using vehicles following certain routes in the streets of a medium-sized city in a developing country.

The principal difference between this work and the related works is the integration in the proposed system of the use of LoRa and its LoRaWAN protocol, with an algorithm that allows the prediction of the transit vehicle location when there is no communication or upon GPS failure. Most of the systems described in the related works do not propose any kind of integration regarding providing a different communication alternative (LPWAN type) and, at the same time, providing a vehicle location prediction option.

The rest of this paper is divided into five more sections. Section 2 presents the literature review; Section 3 presents the materials and methods used in this research; Section 4 shows the main obtained results; Section 5 discusses the results, and, finally, Section 6 presents conclusions and future work.

## 2. Literature Review

In this phase, the methodology of systematic mapping [22] was used, which allows the selection, reading, and classification of the available information through five phases: scope of the investigation, inclusion/exclusion criteria, information search, scheme classification, and results. Details for each phase are presented below.

### 2.1. Scope of the Investigation Phase

In this phase, the objective of the systematic mapping was identified, which was to determine existing types of proposals for transit service monitoring systems, which had a certain type of location prediction through ML. The reasons for performing the systematic mapping were also identified, namely, to identify similar proposals and learn about existing studies related to the use of LoRa, LoRaWANs, and/or ML technology to predict location in a tracking system.

### 2.2. Inclusion/Exclusion Criteria Phase

In this phase, the inclusion or exclusion criteria were determined, classifying them by search or analysis. The inclusion criteria were the year of publication (2017 or later); terms that the title and/or abstract should contain; and the covered topic (related to tracking and tracing prototypes or location prediction algorithms).

### 2.3. Information Search Phase

In this phase, two search strings were built from the keywords obtained in the previous phases. The first search string was: ("machine learning" OR "artificial intelligence") AND ("prediction") AND ("location" OR "tracking") AND ("vehicle"). The second search string was: ("LoRa" OR "LoRaWAN") AND ("vehicle") AND ("location" OR "track-

ing"). These strings were used to query the Scopus and Science Direct databases, yielding 1978 documents.

### 2.4. Scheme Classification Phase

The types of research and the research context were determined with the initial review of the documents (reviewing the abstract and title), by identifying similar characteristics and grouping them together (as indicated in [22]). In this phase, the terms of the two types of classification were established: type of research and research context.

The types of research determined were:

- Evaluation research (this type of work performs an evaluation investigation and does not make a proposal for a solution);
- Solution proposal (this type of work proposes a solution but does not perform measurements);
- Validation research (this type of work performs tests or experiments to validate the proposed solution);
- Others.

The considered contexts were:

- Vehicle tracking;
- Location prediction;
- Transit vehicle tracking;
- Transit vehicle tracking with LoRa or LoRaWANs;
- Vehicle tracking and prediction;
- Vehicle tracking and prediction with ML;
- Discarded.

In total, 119 documents out of the 1978 obtained in the search phase were selected in the scheme classification phase. These documents were classified by type of research and research context.

### 2.5. Results Phase

In this last phase, an intersection of the number of documents with the type of investigation and the context of the investigation was obtained. This information was represented in a bubble map, which is presented in Figure 1.

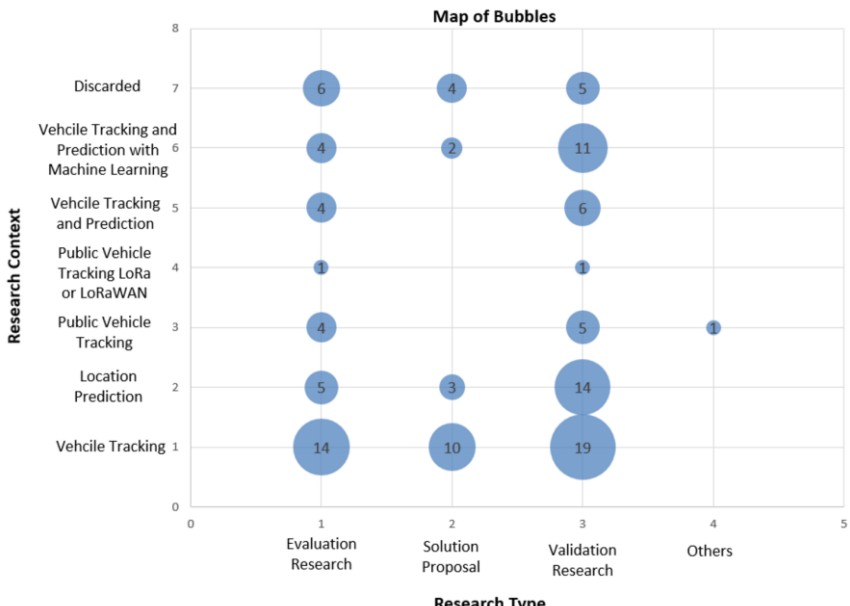

**Figure 1.** Bubble map of systematic mapping.

A total of 100 documents were discarded out of the 119 documents found and represented in the bubble map. The bubbles of the intersections that had a direct relationship with the research work and in which there were not too many previous works were selected. The 17 papers identified in the row of the research context called "Vehicle tracking and prediction with ML" were selected. Additionally, the two works identified in the research context row called "Transit vehicle tracking with LoRa or LoRaWAN" were selected.

Before doing a detailed review of the documents, a new review of other sources was performed to find new related articles. Five new documents related to the use of LoRa and/or LoRaWANs in transit service were found, for a grand total of twenty-four documents.

Based on the above, it was evidenced that no research was found on transit vehicle tracking through the use of LoRa/LoRaWAN technology that also predicts vehicle location using ML models.

### 2.6. Evaluation of Selected Documents with Systematic Mapping

From the twenty-four documents found, nine articles were subsequently discarded through detailed review because they tracked automobiles through unmanned aerial vehicles, cameras, automobiles as vehicular clouds, and the interconnection between them. These methods are not directly related to the scope of this research. Two other articles [23,24] were discarded because they were related to traffic modeling, characterization, and/or forecasting, rather than the tracking and monitoring of transit vehicles.

Table 1 presents a summary of the considered aspects in the proposals of the remaining 13 documents. The proposal of this research takes into account all of the identified points while the related works only take into account some of these aspects.

**Table 1.** Summary of the evaluation of the related works.

| Criteria/Proposal | [2] | [6] | [7] | [8] | [12] | [13] | [14] | [15] | [16] | [17] | [18] | [25] | [26] |
|---|---|---|---|---|---|---|---|---|---|---|---|---|---|
| Use ML models to predict location. | | | | | ✓ | ✓ | ✓ | ✓ | ✓ | ✓ | ✓ | ✓ | |
| Vehicle tracking with Lo-Ra/LoRaWAN. | ✓ | ✓ | ✓ | ✓ | | | | | | | | | ✓ |
| Use of affordable sensors and devices. | ✓ | ✓ | | ✓ | | ✓ | ✓ | | | ✓ | | | ✓ |
| Medium-sized or small-sized city as context. | ✓ | ✓ | ✓ | ✓ | | | | | | | | | ✓ |

In the finally selected 13 works, after performing the filtering proposed by the PRISMA methodology, it was identified that no proposal performed an online tracking process of the transit vehicle in such a way that it integrated a new proposal of communication technology, between the IoV device and the equipment on the road, with a prediction of the vehicle's location. The use of LPWAN communication technologies, instead of conventional technologies (such as cellular communication technologies), is useful to enabling low energy consumption and efficient operation. The prediction of the vehicle's location is necessary if LoRa fails or location data cannot be obtained through a GPS. The integration of these two elements (LPWAN communication and location prediction) is the main novelty of the proposal presented.

Some of the related works focus on the use of LPWAN technology for the tracking service of transit vehicles. These works are analyzed in detail in Section 2.7 to identify the ideal components of the proposed tracking system.

Additionally, other related works focus on the use of computer algorithms to predict the location or trajectory of transit vehicles and facilitate their tracking when a GPS is used and the service is not available or when a location system other than a GPS is used. Section 2.8 analyzes, in detail, the possible algorithms to be used in the proposed system.

### 2.7. Literature Analysis for the Definition of System Components

The ML features (variables) in this work are directly linked to the sensors and devices used to collect vehicle information. Additionally, the work proposes to perform the tracking online; therefore, the storage of the data in the dataset must also be conducted online to make predictions with the data that are being collected. Finally, the problem to be solved must use the data that can be measured in the context of interest. Due to this, for the selection of the features (variables), the sensors, and the devices, an analysis of the identified literature and an evaluation of the context were conducted to determine that it is feasible to measure with the devices and sensors that are available in the problem's context.

Different variables, devices, models, and/or data processing techniques used to improve ML-based real-time vehicle tracking systems were reviewed. The architecture of the transit vehicle tracking service proposed by the ARC-IT (American ITS architecture) was also considered for the definition of the necessary elements for the proposed system [20].

### 2.7.1. Data Collection Device

Microcontroller cards were the most used devices in the works that performed the data collection of vehicular routes. The works [2,6–8,14,17,26] featured the use of Raspberry, Arduino UNO, Arduino Nano, ESP32 LoRa Heltec, and Heltec Cube Cell HTCC–AB02S microcontroller cards.

Refs. [12,13,16,18] did not mention the devices used for data acquisition. Refs. [15,25] used mobile devices and software simulation, respectively. These last two options were not considered because they are not related to the scope of the project, which aims to evaluate the performance of LPWANs in a real environment.

Regarding wireless communication technologies used to transmit the data collected by the devices, refs. [2,6–8,13,14,17,18,26] featured the usage of LoRa, the NBIoT, Bluetooth, Xbee, and/or the GSM/GPRS.

Refs. [6,26] included the usage of the LoRaWAN network protocol in addition to LoRa.

### 2.7.2. Sensors

In documents [2,6–8,12,14–17,26], the GPS was the principal sensor used to determine location. Refs. [12,14,16] also include gyroscope and odometer sensors in addition to the GPS to improve the accuracy of the prediction models. In [13,18], Bluetooth sensors were located at a certain distance from the roads to detect vehicles, thus avoiding the use of a GPS. It is worth noting that when the works are related only to the measurement of location, only the GPS was used; meanwhile, when location prediction was performed in addition to location, additional sensors, such as gyroscope and odometer, were used.

The Bluetooth sensors used in the works [13,18] belong to a network already established in a large city, for multiple purposes. Implementing a similar network would require a larger budget compared to the GPS option.

Considering the above, it is important that the vehicle data collection device has a GPS sensor, an odometer, and a gyroscope.

### 2.7.3. Variables

In the reviewed literature, 27 variables related to the research project were found, with the most common being position (latitude and longitude), speed, and timestamp (time and date). Variables closely related to obtaining the location online and location prediction using ML algorithms were prioritized.

In some reviewed papers [12,14,17], the authors explain the location prediction of a vehicle as the advanced knowledge of the place where such a vehicle will be, with high precision. Geographic coordinates play an important role, as well as the speed or rotation of the vehicle that are used, in reducing calculation errors in the algorithms.

Refs. [7,13,17] present the variable of the arrival time of a vehicle at a specific place (generally a bus stop). This variable is measured because of the output data of the forecasting tools, and also of the sensors used at different points, when alternative monitoring

is complemented with GPS data. Because the sensors are located inside the vehicles and not at the bus stops, arrival time can be substituted for packet arrival time at the data collection platform.

In [14], the distance traveled by the vehicle and its rotation are measured in addition to the position variables; so, when the signal is interrupted (the satellite information is not received by the GPS), the algorithm works with additional information directly related to the location and the path taken, improving prediction accuracy.

Considering the above, the following variables should be measured by the devices and sensors:

- Latitude;
- Longitude;
- Speed;
- Angular speed;
- Traveled distance.

### 2.7.4. Selection of Devices, Sensors, and Variables

The aforementioned variables should be captured by the GPS and IMU sensors. The devices selected to perform the measurements were the following:

- ESP32 LoRa (with built-in LoRa), a GPS module, and an IMU module;
- Cube Cell HTCC–AB02S (with a built-in LoRa and GPS) and an IMU module.

### *2.8. Literature Analysis for the Definition of Algorithm Options*

This section analyzes the use of algorithms in the reviewed literature [12–18,25]. In [12], two types of algorithms were used: Kalman Filter (KF) and Multi-Layer Perceptron (MLP).

In [13], the next stop and arrival time were predicted by two models using hybrid and sequential Long Short-Term Memory (LSTM) networks. The model inputs were the current position, time of day, day of the week, and travel time between the current position and the previous position.

In [14], three prediction models were used: an extended KF, a Neural Network (NN), and an Autoregressive Integrated Moving Average (ARIMA) model. The KF predicted the current position of the vehicle based on the integration of a GPS, an odometer, and gyroscope data. In the absence of a GPS signal, the KF reduced its performance. The NN and ARIMA models were used to reduce extended KF deficiency.

In [15], deep learning was used to predict the trajectory of the vehicle. The selected algorithms to achieve this were a Convolutional Neural Network (CNN) and a Deep Bidirectional Long Short-Term Memory (DBLSTM) network. The data used by the system were the records of the vehicles, which were detected by license plate recognition equipment.

In [16], the input data were the route of an autonomous vehicle equipped with a GPS sensor and a high-precision Inertial Navigation System (INS). The route points generated a road map following a line in the curve of interest; these data were applied to a curved trajectory-tracking algorithm, establishing a relationship between the speed of the vehicle and the fixation point.

In [17], the input data were the positions, provided by a GPS sensor. These data were sent, together with the identification of the vehicle, through the GSM cellular network to a server in the cloud. The system user could monitor a moving vehicle in real-time through a mobile application and a Neural Network (NN) predicted the estimated time of arrival. The NN inputs were the times commonly used to reach the different bus stops. The output of the NN was the predicted time (in seconds) to reach the different bus stops.

Ref. [18] used an Attention-based Recurrent Neural Network (ARNN) algorithm, which used a non-sequential input to predict the trajectory of a vehicle. The attention mechanism served as an interface between the data processed by the RNN and the external data. A sequence of cells of the vehicle trajectory was generated. The output of the algorithm was the probability of reaching each cell.

Ref. [25] used a vehicle trajectory prediction based on a Generative Adversarial Network (GAN). The location of the vehicle was used as input. The road was divided between two adjacent intersections into two road segments (Road Segments or RSGs) with opposite directions. A GAN was used to train a vehicle driving location prediction model on the RSGs. If the vehicle moved to another RSG or reached the end of the current RSG, a turning model was raised. In general, the GAN allowed the researchers to predict the behavior of the driver and the turning model allowed them to predict the trajectory.

## 3. Materials and Methods

The following details how each of the design, development, and test phases of the proposed system were performed. Initially, information on the proposed communication technology is presented. Subsequently, the location prediction algorithm selected for the proposal is presented. Next, the process of designing and developing the prototype of the system is presented. Finally, the design and development of the proposed system tests are presented.

The main difference between this work and previous research is the integration of the proposed system of LoRa with an algorithm that allows the prediction of the location of the transit vehicle. The design and development of the data acquisition device are also other significant differences to the related works since most of them use previously collected datasets. Finally, the tests performed in a context of interest, with vehicles following certain routes in the city, are also a significant differentiator from the other investigations.

### 3.1. LoRa and LoRaWAN Technologies

LoRa is a type of wireless modulation that is used for long-range communications. Its operation is based on a chirp spread spectrum allowing low power consumption, long distances, and robustness against interference [4,5]. LoRa has associated communication parameters, which are spreading factor, bandwidth, and coding rate.

The spreading factor represents the number of symbols sent per bit of information. It uses six possible values (7–12) [27]. Bandwidth represents the range of frequencies used in the transmission band. The bandwidths used are 125, 250, and 500 KHz [27]. Finally, the coding rate allows error correction and detection and protection against interference. The allowed coding rates are 4/5, 4/6, 4/7, and 4/8 [27].

LoRa defines the physical communication layer; meanwhile, the LoRaWAN is the communication network protocol of the system. A LoRaWAN uses a star topology, ideal for low power consumption, allowing single-hop communication between devices and gateways and a long range. In a communication using only LoRa, with several devices operating at the same time, there is a risk of collisions and packet loss due to the non-orthogonal nature of the communication because the spreading factor would have a fixed value. In contrast, using a LoRaWAN allows automatic variation of the spreading factor [5].

Line of sight is crucial in LoRa communications because it uses low-power radio waves that can be blocked or interfered with by physical objects, such as buildings, trees, rough terrain, etc. The ability of the radio signal to penetrate physical obstacles is limited. Therefore, line of sight is important because having fewer obstacles results in greater reliability and a greater range of communication [28].

In LoRa, the terminal device searches for a gateway to use to transmit the data; if no gateway is found, the data are considered lost. On the other hand, in a LoRaWAN, the communication with the gateway has to be previously established (with a login, for security) before sending the information packets; this is necessary to ensure network security and prevent unauthorized devices from injecting data into the network [29].

Figure 2 presents the data transmission using a LoRaWAN in this proposal. The data collection devices communicate directly with the gateways. Then, the gateways send the data packets to the LoRaWAN server; finally, the data packets are sent to the web server. Taking the above into account, each packet travels through three network hops.

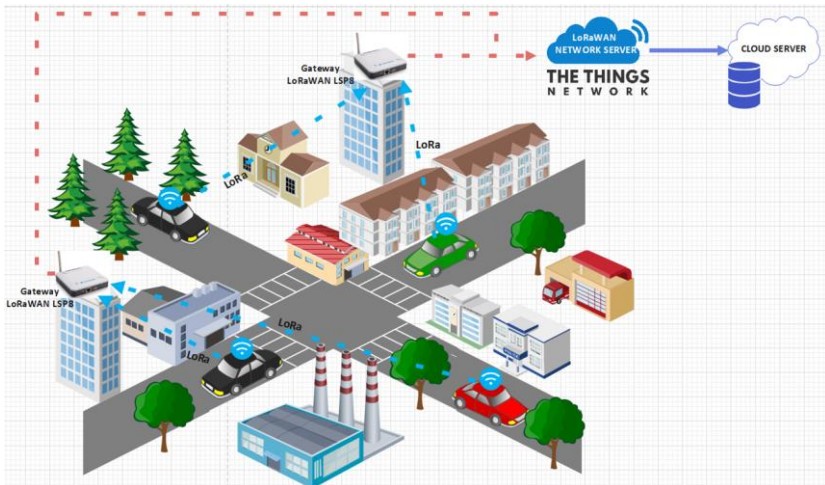

**Figure 2.** Data transmission using a LoRaWAN in the proposal.

### 3.2. Selection of Algorithm for Location Prediction

The algorithms presented in Section 2.8 were compared in detail, reviewing the metrics used in each case. A detailed comparison table can be found in Appendix A. The algorithms compared in Appendix A are those that were used by the related works; they do not correspond to all of the algorithms used in this work.

According to the table, the type of algorithms that obtained the best performance were Artificial Neural Networks (ANNs), with an accuracy greater than 90%. The convolutional neural networks and the DBLSTM network obtained an accuracy of 76%. Sequential LSTM neural networks presented an accuracy of 70.4% and the hybrid LSTM network reached 51.4%.

One aspect that was reviewed was the viability of the implementation of the algorithms. The Python programming language has the Keras library that allows the implementation of a wide variety of neural network models, such as the CNN, NN, RNN, and LSTM. Another important aspect to consider was that the algorithm had been implemented in an experiment close to the context of the application. The experiments performed with sequential LSTM and DBLSTM were the closest to the required context.

Based on the precision values of the previous algorithms and the feasibility of implementation, sequential LSTM and DBLSTM were chosen to develop vehicle location prediction in this project. The operation of the two types of neural networks that were used in this proposal is explained below (Sections 3.2.1 and 3.2.2).

#### 3.2.1. LSTM Network

An important and advanced technology in the field of deep learning is the RNN, which is also the result of developments in artificial neural networks. Variants of the RNN include long short-term memory (LSTM) and the Gated Cyclic Neural Network (GRU). In contrast with feed-forward neural networks, variants of RNNs add other weights to the network. For tasks such as time sequencing, the advantage of adding time series to neural networks is that context information in time sequence prediction can be clearly learned and used [15].

LSTM networks can learn the temporal dependency in sequential data. As LSTM can automatically extract and pass relevant information down the long chain of sequences to make predictions, it is suitable for learning sequential movement patterns in urban trajectory data [13].

The structure of a LSTM model contains an input layer, a hidden layer, and an output layer. The input layer initializes the input data for the subsequent layers while the output layer performs the prediction task [30].

The hidden layer is able to learn features of the input sequence through a recurrent unit called a memory block, unlike an artificial neural network, which has no memory at all. The memory block of the hidden layer contains a cell, which transfers relevant information along the sequence and through three gates (the forgetting gate, the input gate, and the output gate), which regulate the flow of information [30].

### 3.2.2. DBLSTM Network

Recurrent Neural Networks (RNNs) are a class of neural networks that are designed to process sequential data, such as natural language, audio signal, or video. Unlike Convolutional Neural Networks (CNNs), which are mainly used for image classification tasks, RNNs have the ability to capture information from the previous sequence and use it to predict the next sequence. A variant of RNNs is the deep bidirectional LSTM (DBLSTM), which is based on the use of LSTM (long short-term memory) units in a deep architecture. LSTM units are an extension of RNNs that allow neural networks to capture and store long-lived information. DBLSTM uses multiple layers of LSTM units, allowing them to model more complex sequences more accurately than standard RNNs [31].

DBLSTM can be understood as stacking multiple Bi-directional Long Short-Term Memory (BLSTM) layers on top of each other, for example, each layer attempts to model high-level abstractions of a sentence. The first layer operates at character-level input and the last layer makes predictions at the sentence level, e.g., whether it is of positive or negative sentiment [32].

A BLSTM consists of two LSTMs that are run in parallel. The first LSTM processes the input sequence from left to right and the second LSTM processes the input from right to left. At each time step, the hidden state of the BLSTM is the concatenation of the forward and backward hidden states. This initialization allows the hidden state to capture both past and future information that exploit the information that follows both directions [32].

In addition, DBLSTM networks also have the ability to learn hierarchical representations of sequential data, making them especially useful for natural language processing and speech recognition tasks. DBLSTM networks have been successfully used in a variety of applications, such as text processing, speech recognition, and machine translation. They have also been used in the field of computer vision for object tracking and image description generation [31].

### 3.2.3. Initial Tests of the Algorithms

As mentioned in Section 3.2, it was considered convenient to use sequential LSTMs and DBLSTMs in the development of vehicle location prediction. LSTM and DBLSTM are Recurrent Neural Network (RNN) architectures commonly used for sequence prediction tasks, including time series forecasting and natural language processing. These networks are trained using a process called Backpropagation Through Time (BPTT), which is an extension of the backpropagation algorithm designed for sequences. For the development of the algorithm and its respective adjustment and training, the Python language was used, using some ML libraries that allow for implementing the mentioned RNNs. The number of epochs used was 100, the Bayesian optimization method was used, and the number of neurons depended on the optimization algorithm within a range of 10 to 200.

In the ML model used, k-fold cross-validation was not applied since it is not a technique used in models that use RNNs such as those selected for the algorithm (LSTM and DBLSTM). This was verified in related works in which RNNs were also used to make predictions.

To ensure that the LSTM and DBLSTM algorithms were not overfitted, some strategies were employed during the training and evaluation process. The strategies were the following:

- The validation set was used to tune hyperparameters and monitor performance during training;

- A regularization technique, such as recurrent dropout, was applied, helped by adding penalties to the loss function based on the complexity of the model's parameters.

The initial tests were performed in two phases. First, a sequential LSTM network was implemented with the aim of verifying its proper prediction of location (latitude and longitude) and time from certain datasets. Here, a dataset generated in [33] was used to perform location (latitude, longitude) and time predictions.

Then, for the second phase of the initial tests, a dataset generated by the route of a vehicle and the sending of information using LoRa communication and its LoRaWAN protocol was used to perform location and time prediction; but, this was conducted using GPS data, timestamps, and data taken from an IMU device as inputs.

In Appendix B, the results on the accuracy of the prediction algorithm applied to two different datasets are presented. The first dataset consists of data extracted from a database containing GPS-obtained position data in the city of Rio de Janeiro, Brazil [34]. The second dataset was built by the authors of this paper, with the position and inertial data collected on a small road in the city of Popayán, Colombia.

### 3.3. Design and Development of the System Prototype

3.3.1. System Prototype Design

The prototype design of the system was performed based on the identification and selection of the variables, parameters, and devices to be used. In this way, the hardware, software, and communication components that the system should manage were defined.

The prototype design consists of three modules, one for data collection, one for data receiving, and a last one for information processing. Figure 3 shows the system prototype design, identifying each of the modules. This figure was designed using the modules recommended in the ARC-IT's Intelligent Transportation System (ITS) architecture (American architecture widely used worldwide) [20]. The ITS architecture proposes certain modules, actors, and communications for the provision of the transit vehicle tracking service, which are specifically detailed in [19]. Figure 3 identifies the Vehicle-to-Roadside (V2R) communication used in the IoV, in which the device located in the vehicle (IoV device or On Board Equipment, OBE) sends the sensed data to a wireless gateway located near the roads through LoRa.

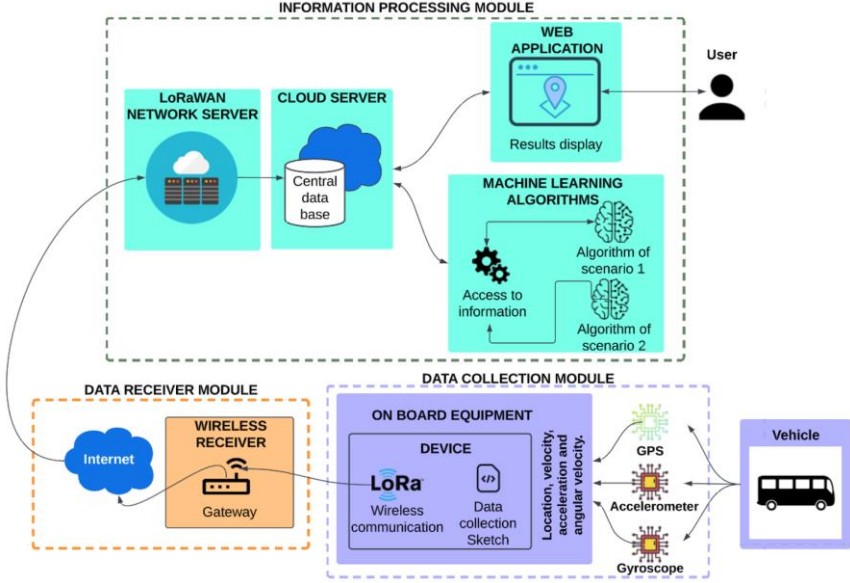

**Figure 3.** Proposed prototype design for the system.

### 3.3.2. Design of the Data Set Structure

From the identification of variables, analysis of alternatives, and their selection, the selected variables for the data set structure were established.

The data collection module must have an associated dataset in which the data transmitted by the on-board device on the vehicle must be stored (this table is named the "On-board device table" and has 17 features or variables). At the same time, there must be a general dataset to store some of the data collected by the on-board device and the data obtained from the output of the ML algorithms; this is required for having a table where the location of the vehicle can be reviewed at any moment (either a real location sent by the on-board device or the location predicted by the algorithm). This table is named the "General data table" and has 17 features or variables. The two mentioned tables (dataset structure) are presented in Appendix C.

### 3.3.3. System Prototype Development

This section specifies each of the electronic devices, technologies, and tools selected to implement the prototype from the design made.

### Development of the Data Collection Module

To implement this module, a microcontroller card with certain integrated modules and some additional components were used.

Two types of microcontroller cards were used due to the low availability of one of them in the market. The two selected cards were the Heltec Cube Cell HTCC-AB02S and the Heltec ESP32 LoRa. The second card required a GPS module, in this case, the Ublox 6M. Both cards required an IMU MPU-6050 module. The schematic diagram of the two devices used is presented in Figures 4 and 5.

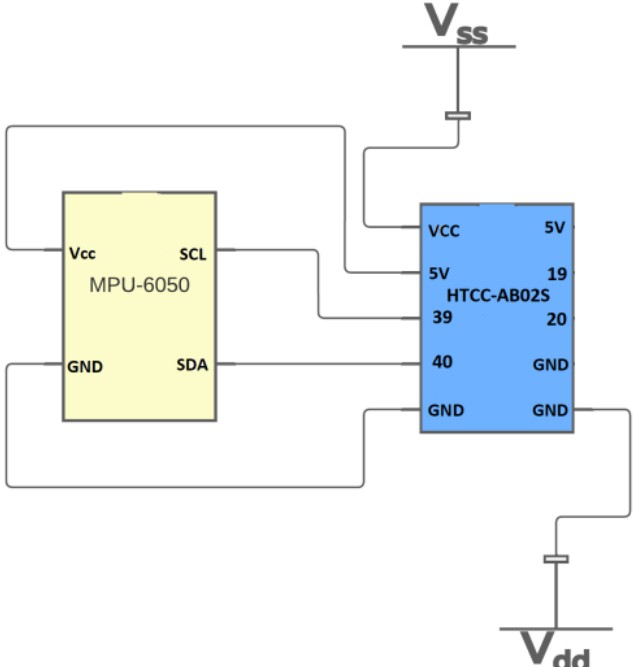

**Figure 4.** Schematic diagram of devices Heltec Cube Cell HTCC-AB02S and MPU-6050.

### Development of the Data Receiver Module

The device chosen for this task was a LoRaWAN gateway. Based on the available equipment in the market and the availability of devices, Dragino's LPS8 Gateway was chosen.

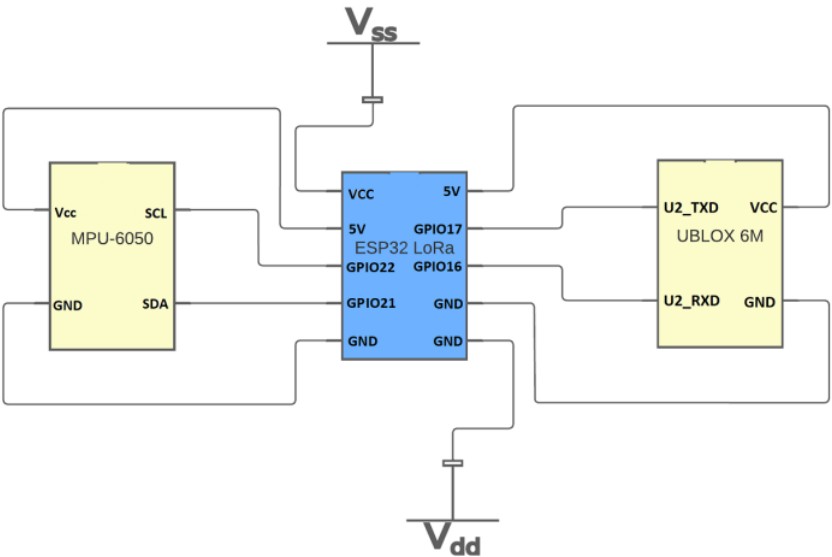

**Figure 5.** Schematic diagram of devices ESP32 LoRa, GPS Ublox 6M, and MPU-6050.

Development of the Information Processing Module

This module, as presented in the prototype design (Figure 3), is made up of 4 components: a LoRaWAN network server, a cloud server, ML algorithms, and web application.

The LoRaWAN network server forwards the data coming from the LoRaWAN gateway. The platform chosen to use to perform this task was The Things Network (TTN) [35]. This tool was selected after reviewing the functionalities of similar platforms in the market and other previous works.

The cloud server is responsible for storing the information from the LoRaWAN server in a database. To perform this function, the service of the Hostinger provider was chosen, considering the costs and technical and operational benefits of various platforms in the market [36]. This server provides web hosting services, database storage services, and management through tools such as phpMyAdmin (free software tool written in PHP, intended to handle the administration of MySQL over the Web).

The selected ML algorithms, as presented in Section 3.2, were LSTM networks. A general architecture of the prediction system was conceived from the identification of the objectives of the ML algorithms. This architecture is presented in Figure 6. The execution flow of the architecture is as follows. The data monitoring block accesses the information stored in the database of the server in the cloud. From the monitoring of the data, the blocks of the execution of training and execution of scenarios come into operation. When there are enough records in the database, the training component of the model is activated to later generate a trained model that will serve as input for the prediction scenarios. When the monitoring block detects any of the events that make it necessary to execute the prediction algorithms of Scenario 1 or Scenario 2, it proceeds to execute them. With the model already trained, each algorithm generates the respective outputs of each scenario to later store these results in the database.

A web application was developed, for visualizing, in an adequate way, the route monitoring and the prediction of the future locations of the vehicle.

The web application was developed using the PHP language and a Google Maps API. The first component of the application receives and stores the data from the TTN platform while the second component allows viewing and filtering the data.

In the second component, three options are presented. The first option, *Routes*, allows access to the routes to be performed in the execution of the system's functional tests. The second option, *Vehicles*, allows you to select the vehicles registered in the system and see their characteristics: assigned route, location of gateways, and name of the onboard device. Finally, the third option, *Tracking*, is presented; it allows you to filter the data to be displayed from a vehicle in a time interval.

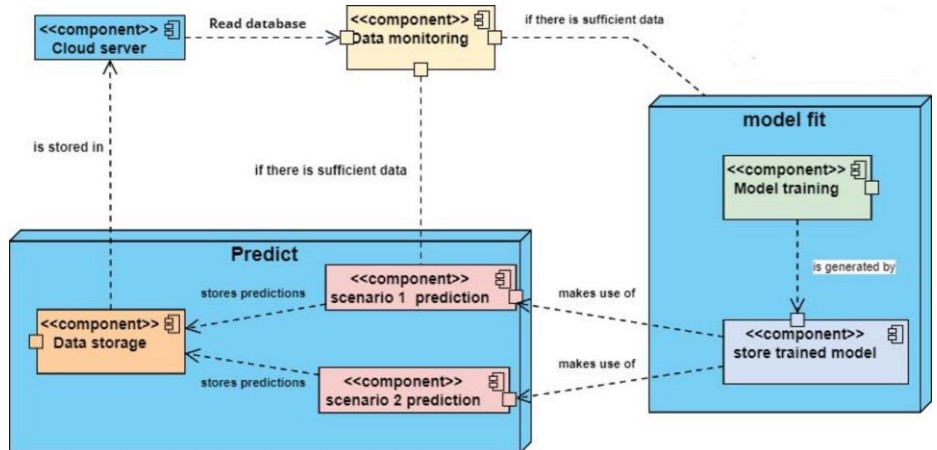

**Figure 6.** Algorithm architecture of the general prediction system.

*3.4. Design of the Field Tests and Construction of the Data Set*

The field tests were designed with the objectives of evaluating vehicle tracking through LoRa communication technology and the LoRaWAN protocol and evaluating the performance of the ML algorithm in predicting future locations when LoRa communication or GPS connection is lost. To meet the above objectives, two types of tests were defined: controlled and uncontrolled. Controlled tests were designed to collect route location data, train the ML model, and observe the behavior of the prediction in Scenarios 1 and 2. The uncontrolled tests were designed to check the performance of the prediction algorithm in a real environment, where two previous scenarios might occur.

3.4.1. Definition of Routes and Test Elements

The tests were performed considering the following:

*IoV devices used:* Four data collection devices were used, three using the Heltec Cube Cell HTCC-AB02S card and another one using the Heltec ESP32 LoRa card.

*Quantity and type of vehicles used*: Four cars were used. Although the objective of the work was to monitor the "collective" transit service, private cars were used for testing because it was not possible to obtain transit vehicles. For the controlled tests, a single vehicle would travel the 3 presented routes. For uncontrolled tests, Vehicles 1 and 2 would take different trips along Route 1 while Vehicles 3 and 4 would, respectively, take Routes 2 and 3.

*Number and location of gateways*: It was possible to obtain two Dragino LPS8 Gateway gateways for the development of the tests of this work. The gateways were located at a height of approximately 15 m from the roads along which the routes were made; the latitude and longitude locations of each one were as follows: Gateway 1: 2.449317, −76.626888, Gateway 2: 2.448446, −76.621751.

*Characteristics of the scenarios selected for the tests:* The tests were performed in the city of *Popayán*, a medium-sized city in Colombia (a developing country in South America). The city of *Popayán* currently has approximately 400,000 inhabitants.

Two types of tests were determined, controlled and uncontrolled. The controlled tests were named in this way because the moments in which there were communication problems were controlled, that is, such events were forced to show what happened in each of the two scenarios considered as communication problems (the two possible scenarios that are explained in Point 2 of Appendix B). In the uncontrolled tests, no communication loss event was forced, the system was simply allowed to operate normally and when a communication problem was detected, the respective algorithm was executed to make the corresponding prediction.

The type of streets on which the tests were performed are common in the city of *Popayán*; they are two-lane streets. The average speed on these city streets is approximately



20 km/h. The private vehicles used for the tests traveled the streets with the presence of traffic congestion and flowed free at times with a medium level of traffic (between 10:00 a.m. and 5:00 p.m.). The streets where the test routes were defined have traffic lights at some of the street junctions, which is common on public transport routes in the city.

*Routes used*: Three routes were defined for the tests. These routes are presented in Figures 7–9. The indicated points in these figures (A, B, C, D, E, . . ., etc.) do not correspond to stopping points of the routes (or stations); they are only geographical points taken as reference to make it easier to trace the route.

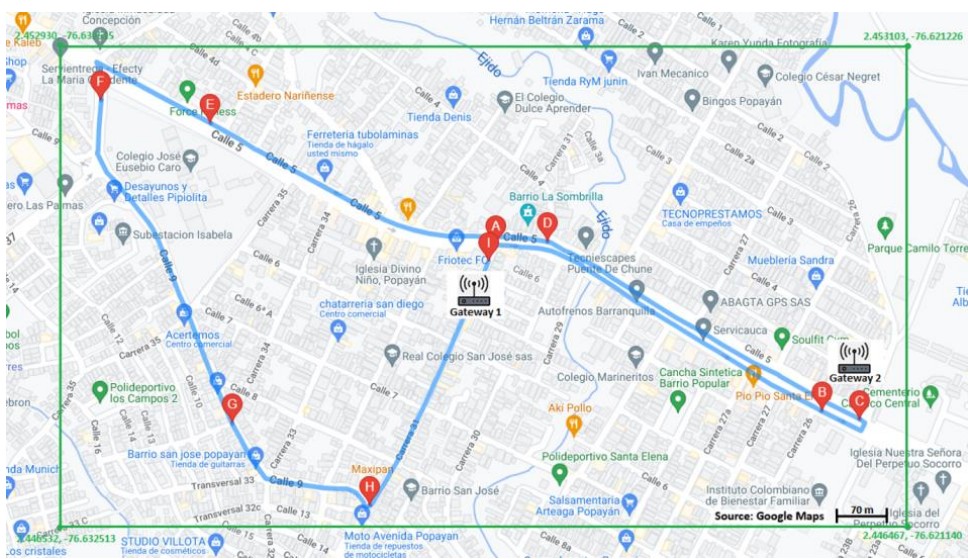

**Figure 7.** Map of Route 1 and locations of the 2 gateways (map taken from Google Maps).

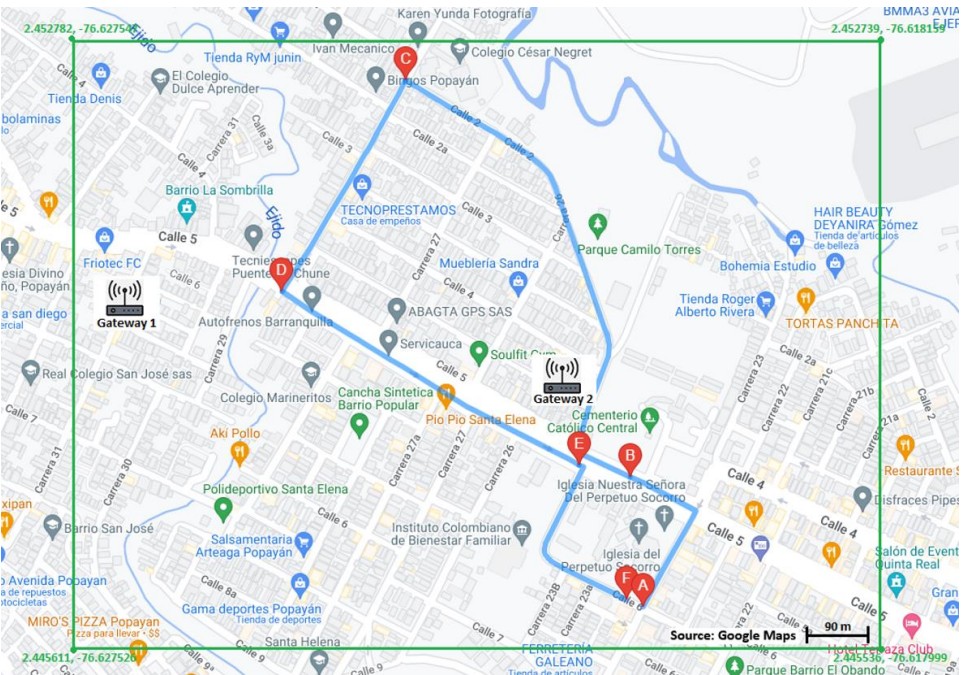

**Figure 8.** Map of Route 2 and locations of the 2 gateways (map taken from Google Maps).

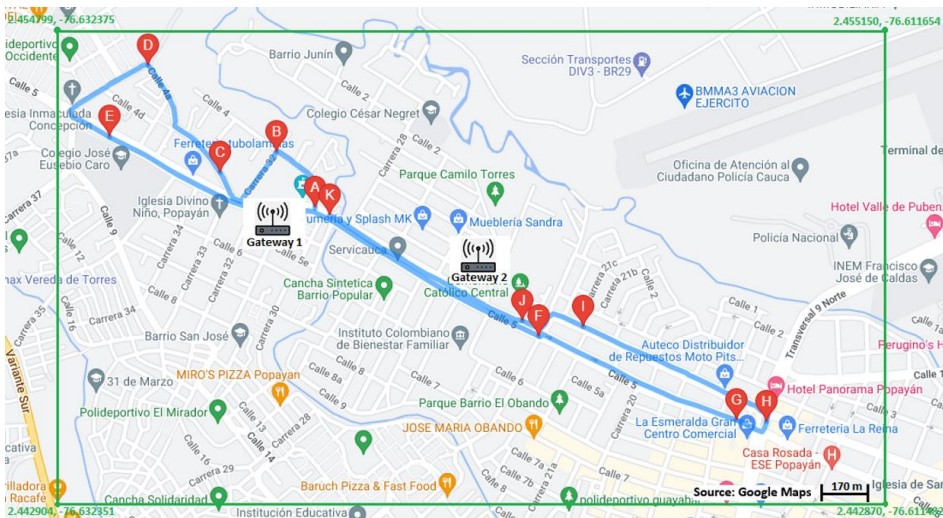

**Figure 9.** Map of Route 3 and locations of the 2 gateways (map taken from Google Maps).

*Vehicle trips:* Based on the previously defined routes, the respective vehicles in charge of traveling each of them were assigned a starting and ending location.

For the controlled tests, a single vehicle would travel the 3 presented routes. For uncontrolled tests, Vehicles 1 and 2 would take different trips along Route 1 while Vehicles 3 and 4 would, respectively, take Routes 2 and 3.

### 3.4.2. Controlled Test Design

The design of the controlled tests was aimed at collecting enough data for the training algorithm and simulating the absence of a GPS signal and/or LoRa connection.

Based on the above, the collection of 300 data points for Routes 1 and 3, and 200 data points for Route 2, was planned for generating a model through the training algorithm and allowing us to execute the uncontrolled field tests for the three described routes.

The data collection was planned to be performed with a single vehicle, with a single driver, to collect a total of 300 data points (Routes 1 and 3) in an approximate time of 65 min for each route and 200 data points (Route 2) in a time of 43 min.

Uncontrolled tests were planned for the two types of scenarios (Scenario 1, in which the device is not able to send LoRa packets to the gateway, and Scenario 2, in which the GPS system is not able to capture the data location data and only IMU measurements are sent).

With the objective of obtaining the real location of the vehicles during the execution of the prediction scenarios, a smartphone application was developed to store the data obtained through the phone's GPS sensor in a local database. Then, the stored data were used to compare it with predictions from the system, allowing us to evaluate the accuracy of the prediction.

### 3.4.3. Uncontrolled Test Design

Uncontrolled tests were planned to observe the behavior of the LoRa connection in vehicle tracking and the location prediction (latitude and longitude) of the vehicle in the absence of this communication. To perform this test, the already-trained model was used. In each run, the algorithm would calculate the time difference between the current time and the time of the last data point recorded; when the difference was greater than 13 s (average time between data received during training), the algorithms would generate the possible future locations of the vehicle until receiving a new packet with real location information from the server.

For these tests, Vehicles 1 and 2 were allocated to travel Route 1, with Vehicle 1 approaching Gateway 1 and Vehicle 2 moving away from it. At the same time, Vehicles 3 and 4 would travel Routes 2 and 3, respectively.

All of the vehicles should start and end their trip at the same time. Once the longest route (Route 3) was completed 3 times, the prediction algorithm would be executed for Vehicles 2, 3, and 4. Prediction was not executed for all vehicles because previous performance tests showed that the maximum number of connections per hour (500 connections) to the database was reached 5 min after execution in all four vehicles was started.

Table 2 summarizes the schedule of the uncontrolled test plan. The expected number of records in Table 2 was obtained from the estimated duration of the test (1 h 42 min) and the average time between the received data packets, which was 13 s under ideal conditions (line of sight between the transmitter and receiver).

**Table 2.** List of planned routes for uncontrolled tests.

| Vehicle | Route | Scheduled Date | Scheduled Start Time | Scheduled End Time | Expected Number of Records |
|---------|-------|----------------|----------------------|--------------------|----------------------------|
| 1 | 1 | 11 December 2022 | 11:55 a.m. | 1:37 p.m. | 475 |
| 2 | 1 | 11 December 2022 | 11:55 a.m. | 1:37 p.m. | 475 |
| 3 | 2 | 11 December 2022 | 11:55 a.m. | 1:37 p.m. | 475 |
| 4 | 3 | 11 December 2022 | 11:55 a.m. | 1:37 p.m. | 475 |

As previously mentioned, a smartphone application was used to obtain the real locations of the vehicles during the execution of the prediction scenarios.

## 4. Results

This section presents the results of the tests (controlled in Section 4.1 and uncontrolled in Section 4.2) designed for the prototype of the transit vehicle tracking system, presented in Section 3.4. Additionally, Section 4.3 presents the basic information of the generated datasets, which can be expanded in [37]. Three datasets were generated, one of them for the data collected from the initial training, another for the data collected from the controlled tests, and finally another for the data from the uncontrolled tests. Each of the datasets has two tables in which the data were collected. The explanation of the content of the tables is found in Section 3.3.2 and the structure of each of the two tables is presented in Appendix C.

It should be noted that this work presents the results of two types of tests: the controlled ones, in which the occurrence of some event is forced (such as the loss of LoRa communication or the loss of communication with a GPS), and uncontrolled tests, in which an attempt is made to replicate a real transit vehicle tracking scenario. In contrast, the related works focus mainly on controlled type tests to guarantee that the events to be studied occur. By forcing the events, for a certain time, the real environment is not being reflected; service interruptions are sometimes very short and random.

Another noteworthy aspect of the results obtained in this work is related to obtaining a prediction of specific location data (with latitude and longitude), which can be compared with the actual location data obtained through an application, e.g., on a smartphone located in the vehicle. The difference (in a unit of length, such as meters) between the actual location and the predicted location is easily obtained in this work; meanwhile, in most related works, a calculation of a specific location prediction is not made; a trajectory is calculated so it is not possible to find an exact measure of error.

The type of ML problem solved in this work is regression. The goal of the ML algorithm is to predict the real-time location of transit vehicles (when certain events occur that affect vehicle tracking), which involves estimating continuous numerical values (latitude and longitude coordinates) based on various input data sources, such as the GPS-LoRa module card and Inertial Measurement Unit (IMU). This falls under the domain of regression as the aim is to predict a continuous and quantitative output.

In the context of a regression problem like the one described here, several performance metrics can be used to evaluate the accuracy and effectiveness of the location prediction

model. Mean Absolute Error (MAE) was selected as the best option because the work required the calculation of a particular location (exact latitude and longitude), rather than an approximation or trajectory, as is proposed in most related works. MAE is defined as the average of the absolute differences between the predicted and actual location values.

Sections 4.1 and 4.2 present the general results of the tests. The general objective of the tracking system is to perform the continuous monitoring of the vehicle through a route and determine if such monitoring is achieved with an adequate level of error. In Section 4.1, the results focus on presenting how much data were obtained normally (by measurement) and how much was obtained through a prediction, performing it in the two types of possible scenarios (that is, the two possible events that affect vehicle tracking). In Section 4.2 the results focus on determining what percentage of the route could be tracked with the communications system (LoRa) and what percentage could be tracked with the prediction algorithm. Additionally, information is provided regarding the errors measured between the actual location value (obtained using a smartphone inside the vehicle) and the location predicted by the ML algorithm.

### 4.1. Results of Controlled Tests

Table 3 shows the obtained results in the data collection for the training of the ML model.

**Table 3.** Results obtained from the controlled tests executed for training the algorithm.

| Vehicle | Route | Date | Start Time | End Time | Number of Laps per Route | Obtained Number of Records |
|---|---|---|---|---|---|---|
| 2 | 1 | 2 December 2022 | 10:26 a.m. | 12:20 p.m. | 7 | 302 |
| 2 | 2 | 2 December 2022 | 2:03 p.m. | 2:53 p.m. | 6 | 215 |
| 2 | 3 | 2 December 2022 | 3:38 p.m. | 5:10 p.m. | 4 | 304 |

Table 4 shows the obtained results in the controlled tests forcing the execution of Scenario 1. The tests were not executed on the date they were planned due to failures in the Internet access point in one of the gateways. Initially, the tests were performed on 23 December; but, due to poor weather, they were completed on 27 December.

**Table 4.** Results obtained from controlled tests for Scenario 1.

| Vehicle | Route | Date | Start Time | End Time | Obtained Number of Records | Number of Predicted Data |
|---|---|---|---|---|---|---|
| 2 | 1 | 23 December 2022 | 2:51 p.m. | 3:03 p.m. | 31 | 25 |
| 2 | 2 | 23 December 2022 | 3:14 p.m. | 3:29 p.m. | 33 | 27 |
| 2 | 3 | 27 December 2022 | 9:41 p.m. | 9:58 p.m. | 33 | 44 |

Table 5 shows the results obtained from the controlled tests forcing the execution of Scenario 2. The tests were not executed on the date they were planned due to failures in the Internet access point in one of the gateways. Initially, the tests were performed on 23 December; but, due to poor weather, they were completed on 27 December.

**Table 5.** Results obtained from controlled tests for Scenario 2.

| Vehicle | Route | Date | Start Time | End Time | Obtained Number of Records | Number of Predicted Data |
|---------|-------|------|-----------|----------|---------------------------|--------------------------|
| 2 | 1 | 27 December 2022 | 10:09 a.m. | 10:18 a.m. | 32 | 13 |
| 2 | 2 | 27 December 2022 | 10:27 a.m. | 10:37 a.m. | 33 | 12 |
| 2 | 3 | 27 December 2022 | 10:46 a.m. | 10:58 a.m. | 32 | 11 |

Data points recorded in the controlled tests were plotted on a map. The location data points received from the vehicles are shown in blue and the predicted locations in the absence of LoRa communication are shown in green. Figure 10 shows an example of the maps made for each vehicle, in each of the scenarios.

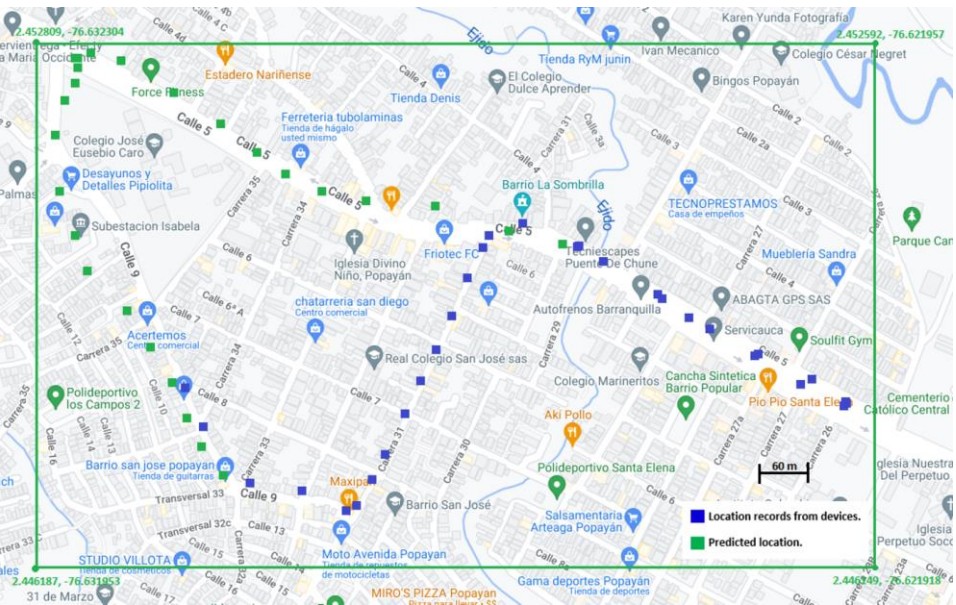

**Figure 10.** Records on the map (controlled test) of Scenario 1 for Route 1 (map taken from Google Maps).

The smartphone application used to obtain the real position worked properly during the tests. From the obtained real data, the coordinates corresponding to each predicted location were extracted, according to the time in which that location was predicted. In this way, it was possible to calculate the difference, in meters, of the actual location at a certain time, with respect to the closest predicted location at that time.

Figure 10 shows the route of Vehicle 2, along Route 1. The device sent 31 records (blue points); whereas, the predicted number of records was 25 (green points). The average error (comparing the real location from a smartphone with the predicted location) was 168 m.

*4.2. Uncontrolled Test Results*

The prediction tests were run for Vehicles 2, 3, and 4 at the time. For all trips, the prediction was running for 25 min. During this time, the connection to the database remained stable.

For these tests, Route 1 was traveled six times by Vehicles 1 and 2, Route 3 was traveled by Vehicle 4 a total of three times, and Route 2 was traveled by Vehicle 3 a total of ten times. Table 6 shows the number of records obtained in the uncontrolled tests.

**Table 6.** Results obtained from the uncontrolled tests.

| Vehicle | Route | Date | Start Time | End Time | Obtained Number of Records | Expected Number of Records | Packet Loss (%) | Coverage (%) | Number of Predicted Data |
|---|---|---|---|---|---|---|---|---|---|
| 1 | 1 | 11 December 2022 | 11:55 a.m. | 1:37 p.m. | 353 | 475 | 25.7 | 74.3 | 0 |
| 2 | 1 | 11 December 2022 | 11:55 a.m. | 1:37 p.m. | 374 | 475 | 21.3 | 78.7 | 18 |
| 3 | 2 | 11 December 2022 | 11:55 a.m. | 1:37 p.m. | 248 | 475 | 47.8 | 52.2 | 30 |
| 4 | 3 | 11 December 2022 | 11:55 a.m. | 1:37 p.m. | 173 | 475 | 63.6 | 36.4 | 42 |

From Table 6, a coverage of 60.4% was obtained, which corresponds to the average coverage of all routes covered with only two gateways and four mobile devices sending packets at the same time. Table 6 also shows that prediction was not performed for Vehicle 1 due to limitations in the cloud server.

Data points recorded in the uncontrolled tests were plotted on a map. Location data points received from the vehicles are shown in blue and predicted locations in the absence of LoRa communication are shown in green. Figure 11 shows an example of the obtained maps for each vehicle (no scenario was forced for this type of test).

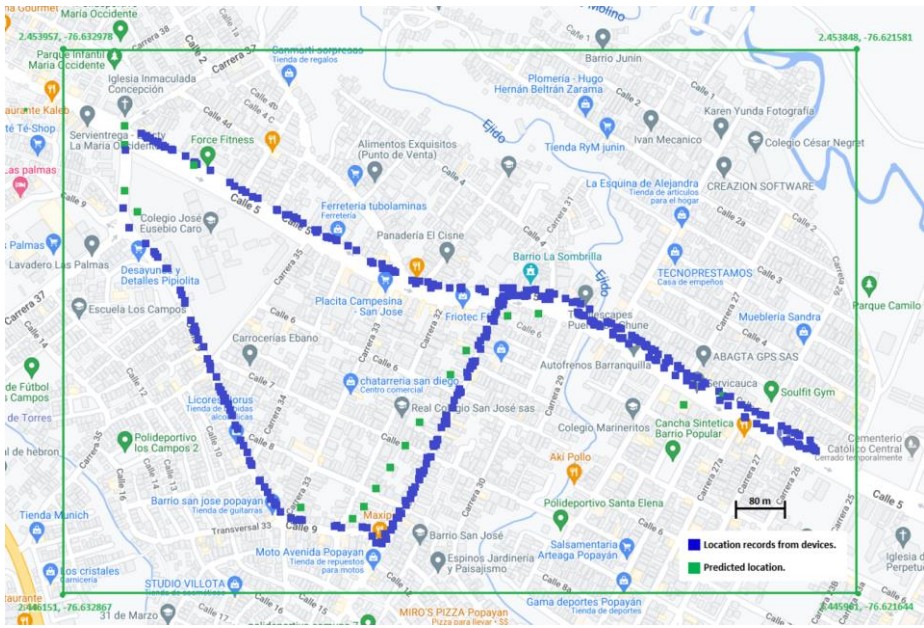

**Figure 11.** Records on the map of the route of Vehicle 2, uncontrolled test (map taken from Google Maps).

The Smartphone application used to obtain the actual position also worked adequately during the uncontrolled tests.

Figure 11 shows the trip of Vehicle 2 (one of the four vehicles that performed uncontrolled tests) along Route 1. The device sent 374 records; whereas, the number of predicted records was 18. The average error (comparing the actual and predicted locations) was 87 m.

For Vehicle 3 (following Route 2), 248 records were sent by the in-vehicle device, there were 30 predicted records, and the mean error was 246 m. For Vehicle 4 (following Route 3), 173 records were sent by the in-vehicle device, there were 42 predicted records, and the mean error was 198 m.

The average error (between the actual and the predicted location) in all of the predictions of the controlled tests (considering the three routes in which the algorithm was executed) was approximately 177 m.

### 4.3. Generated Datasets

The collected information in the performed tests (controlled and uncontrolled tests described in Sections 4.1 and 4.2) was stored in three datasets in CSV format. The first one contains the location data points collected along the three traveled routes to train the models responsible for predicting the location. The second one contains the data points obtained during the execution of the controlled tests. The third one contains the information obtained during the execution of the uncontrolled tests. These datasets are in [37].

## 5. Discussion

It is important to highlight that the use of LPWAN communications technology (such as LoRa) was integrated into the prototype of the developed system with the use of an algorithm that predicts the specific location of the vehicle. This integration was not found in any of the related works. In addition to the achieved integration, it was very important to obtain location-specific predicted data; in this way, it was possible to obtain an error value (measured in distance), which was not found in any of the related works.

Having managed to develop a module of the prototype of the system in charge of detecting events that affect the adequate tracking of the vehicle, when these events occurred, the appropriate algorithms were executed to obtain a specific predicted location. This event detection and integration with the algorithms was not found in the related works either.

One of the specific objectives of the work was to analyze the efficiency and effectiveness of the transit vehicle tracking system, considering the results obtained from the designed tests. Regarding the effectiveness of the system (using LoRaWAN and ML prediction), it can be affirmed that the objective of maintaining continuous tracking of the vehicle that is close enough to the real location was achieved. This was achieved using low power consumption, which is ideal in fleets of electric or hybrid transit vehicles. In addition, by having a system based on an international reference ITS architecture (such as ARC-IT), the development of mobility services for the city is facilitated, which can interact without major inconvenience.

It is also important to mention that one of the most important novelties of our work is to have field tests of the developed prototype. Several related works were found in which no field tests were performed, only simulations were carried out. By presenting the data only in simulation, many important factors are not considered, such as line of sight (LoS) issues, data storage delays, potential GPS coverage issues, and potential packet collision issues.

Next, a discussion is presented regarding the results obtained in the controlled tests and, later, the results of the uncontrolled tests are discussed.

The results of controlled tests show that, despite having considerable separation distances between each of the actual and predicted locations, the predicted data points stay on the previously defined routes (or very close to them). This was evidenced in Scenario 1 where, due to the absence of LoRa communication, the predictions followed the route.

In addition, the predictions for Scenario 2, due to the absence of a GPS connection (controlled tests), showed that the predicted data points were located at the same point. This type of prediction occurred because when collecting the data for algorithm training, the vehicle had to stop at the traffic lights on each of the routes.

When analyzing the errors obtained between the actual and predicted locations in the controlled tests, it is concluded that these significant distances between the geographical points may be due to the lack of synchronization in the system. When there is a lack of LoRa or GPS connection, the respective prediction algorithms of the system are executed and the predicted data are sent continuously without a time interval between each of these data points.

Regarding the uncontrolled tests, it was evidenced that the mean square error obtained for Scenario 1 compared to the controlled tests was better. This is because, in these tests, the interruptions due to the absence of LoRa communication lasted a short time and the predicted data considered data sent by the device and predicted data. In the controlled tests,

there was a larger number of locations with predicted values because LoRa communication was not present for a longer time.

The distance difference between actual and predicted values in uncontrolled tests might also be due to a lack of system synchronization. In the uncontrolled tests, each predicted data point stayed within the established path (or very close to it).

The percentage of packets received at the gateway in the uncontrolled tests, compared to those sent by the devices, was 60.4%, which is relatively high; furthermore, for the remaining 39.6%, the option of prediction worked very well, managing to deliver a location close enough to the real one for the whole route.

The average precision obtained in the uncontrolled tests was 177 m. This value is the difference between the actual location and the predicted location, it is relatively low for a public transportation system. This occurs because, in terms of vehicle travel time, such a distance would represent only a few seconds (approximately 10 to 15 s, depending on travel speed).

Additionally, with the results of the uncontrolled tests, it was possible to demonstrate that the system operates properly when there is more than one vehicle operating, which would happen in a transit vehicle tracking system in which there are several vehicles operating in the same area of the city at the same time.

Regarding the limitation of the number of algorithms that can be executed at the same time, it is important to mention that this could be solved by improving the capacity of the cloud server used for this task.

## 6. Conclusions

Analyzing the obtained results, it can be affirmed that it is possible to improve the characteristics of the current monitoring systems of the collective transit service (and future fleets, considering the trend of use of electric or hybrid vehicles) through the development of a prototype of a system for monitoring the service in real-time, based on IoV, ITS, and ML.

The principal contribution of this work lies in providing an option for a transit vehicle tracking system, applicable in medium-sized cities in developing countries, to improve fleet control and improve user information. Furthermore, it guarantees the efficiency of the energy consumption of the device in the vehicle, reducing the probability of the unavailability of the monitoring service and maintaining a low level of error in the event of data communication failures. Additionally, other important contributions of the work are the datasets generated with the tests performed in a context of interest and the algorithm in charge of detecting problems in the vehicle tracking and generating a prediction, with a good level of accuracy, for the location of the vehicle.

In a transit vehicle tracking system, it is advisable to use LoRa technology with its LoRaWAN protocol to achieve greater security and better range distances and to be able to operate various vehicles for each existing gateway. The adoption of LoRaWAN technology to replace cellular communication (GSM, LTE) would generate considerable savings in the operation. In addition, there are advantages of sustainability, such as the low power consumption of the devices located in the vehicles and the reduction of the required communications equipment in the field.

The LSTM and bidirectional LSTM neural networks showed similar performance when executing location predictions and time predictions in the transit vehicle tracking system prototype. So, it was decided upon to establish a LSTM network for location prediction and a bidirectional LSTM network for timestamp prediction.

By observing the predicted points in the different routes, it is possible to visualize that these points are on the route and not randomly scattered over the real point. Part of the error might be due to a problem with the synchronization of the predicted time with the predicted locations.

The coverage (60.4%) and precision values (177 m of average error) obtained in the final field tests of the prototype were satisfactory, considering that the number of LoRaWAN gateways used in the test was relatively low (only two).

Regarding the restrictions of the developed prototype, the server in which the tracking application was hosted presented a low number of connections per hour; this influenced the interval during which it was possible to execute the prediction algorithms for several routes at the same time. This also influenced the number of routes over which the prediction could be applied.

Based on the development of the prototype and the presented results, the following future works are proposed:

- Modifying the characteristics of the algorithm of the monitoring system to improve the precision of the predictions. Such characteristics can be the hyper-parameters of the model, the size of the training data set, or the number of layers of the LSTM network;
- Modifying the training of the prediction model, making it change as the position information is added to the database, and checking the effects of this change in training on the precision of the obtained results;
- Improving the operational characteristics of the server in the cloud where the prediction algorithms are hosted to remove or raise the limitation on the number of vehicles that can be tracked at the same time;
- Expanding the amount of resources used in system operation tests by increasing the number of gateways, the number of tracking devices, and the number of vehicles. Additionally, increasing routes and travel times;
- Designing and developing new mobility services for the city, using the ITS architectures provided by the ARC-IT as a reference, seeking the interoperability and standardization of these types of services;
- Implementing new functionalities for the IoV device that is used in the proposed system. Considering that it has the ability to sense other vehicle variables and send them through the same means of communication proposed.

**Author Contributions:** Conceptualization, R.S.-C. and Á.P.d.l.C.; methodology, R.S.-C., Á.P.d.l.C. and J.M.M.M.; software, C.A.S.D. and A.S.D.L.; validation, R.S.-C., C.A.S.D. and A.S.D.L.; formal analysis, R.S.-C., C.A.S.D., A.S.D.L. and Á.P.d.l.C.; investigation, R.S.-C., C.A.S.D. and A.S.D.L.; resources, Á.P.d.l.C. and J.M.M.M.; writing—original draft preparation, R.S.-C., C.A.S.D. and A.S.D.L.; writing—review and editing, R.S.-C., Á.P.d.l.C. and J.M.M.M.; supervision, R.S.-C., Á.P.d.l.C. and J.M.M.M.; funding acquisition, Á.P.d.l.C. and J.M.M.M. All authors have read and agreed to the published version of the manuscript.

**Funding:** This research received no external funding.

**Data Availability Statement:** The data presented in this study are openly available in Kaggle at https://doi.org/10.34740/kaggle/dsv/4893922 [37].

**Acknowledgments:** The authors wish to thank Universidad del Cauca (Telematics Department) and Universidad Icesi (ICT Department) for supporting this research.

**Conflicts of Interest:** The authors declare no conflict of interest.

## Appendix A

The algorithms mentioned in Section 3.2 were compared in detail, reviewing the metrics used in each case. It is clarified that the algorithms compared in this Appendix correspond to those used by the authors of the related works. A detailed comparison table can be found at: https://drive.google.com/file/d/1t_XtVUzlTCwFUGiUNBJ8ClgI5oiYivUe/view accessed on 13 September 2023.

## Appendix B

This appendix presents the results on the accuracy of the prediction, applied to two different datasets. The first dataset consists of data extracted from a database on the Kaggle platform, which contains position data obtained by a GPS in the city of Rio de Janeiro, Brazil. The second dataset was built by the authors of this paper with position data and inertial data collected on a small road in the city of Popayán, Colombia. The evaluation

was conducted by comparing the predicted locations with a part of the data that was not included in the data input into the training algorithm.

*Appendix B.1. Initial Tests Using an Existing Dataset*

Initially, there was no dataset to observe the operation of the algorithm; so, a search for a suitable one was performed on a platform called Kaggle [33]. In this platform, a dataset with the name "GPS data from Rio de Janeiro buses" [34] was found; these data were collected during 56 days and they contain the following columns: date, time, order, line, latitude, longitude, and speed. The dataset contained too many records (59,183,745); for that reason, the route of a single bus identified with the number C51623 was selected. This reduced the size of the dataset to 2500 data points from such vehicles.

The first part of the test consisted of predicting the location points (latitude and longitude) of the C51623 vehicle to observe the performance of the algorithm, comparing predicted values to real values. To achieve this, a sequential LSTM network was implemented, where the algorithm inputs were latitude and longitude. A total of 2314 data points out of the 2500 from the selected vehicle were used to train the algorithm. From these records, blocks of 50 consecutive data points were taken to form the input vector that allowed the training of the algorithm. The remaining 186 data points were used to validate that the output generated by the LSTM network was close to the real data.

Figures A1 and A2 represent the results of predicting the location (latitude and longitude) of the vehicle from historical data. The horizontal axis is in dimensionless units and represents the number of generated measurements (a consecutive ID that indicates the number of the measurement) while the vertical axis shows the values of latitude and longitude, expressed in decimal degrees. In Figures A1 and A2, the first 50 validation values were not plotted because there was no correspondence with the predicted data. For this reason, the graphs go up to measurement number 136.

When comparing the prediction and actual value graphs (in Figures A1 and A2), it was observed that the predicted data were close to the actual data. The mean error for the latitude in this test was 0.0045336 and the mean error for the longitude value was 0.0021285 (normalized data). Entering this difference in a "haver-sine" formula [38], which calculates the distance between two geographical points, an average distance of 0.557 Km was obtained. Although this could be considered a large value, the trend of the route is maintained, indicating that the algorithm may be viable for predicting locations.

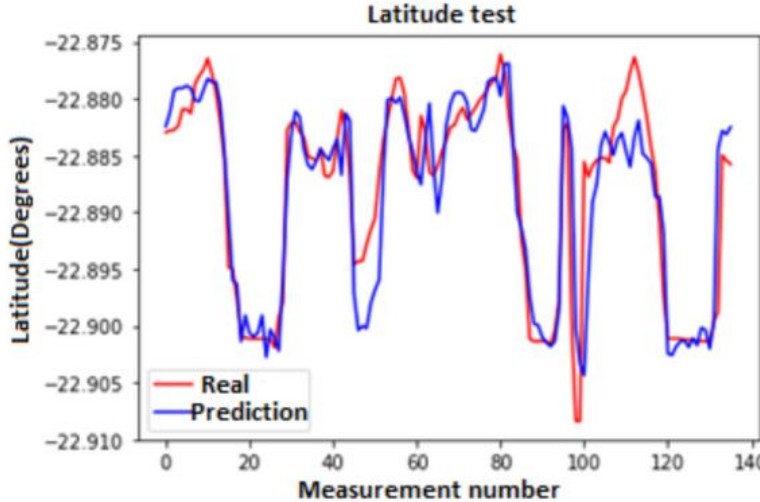

**Figure A1.** Latitude prediction from historical data for a route in Rio de Janeiro.

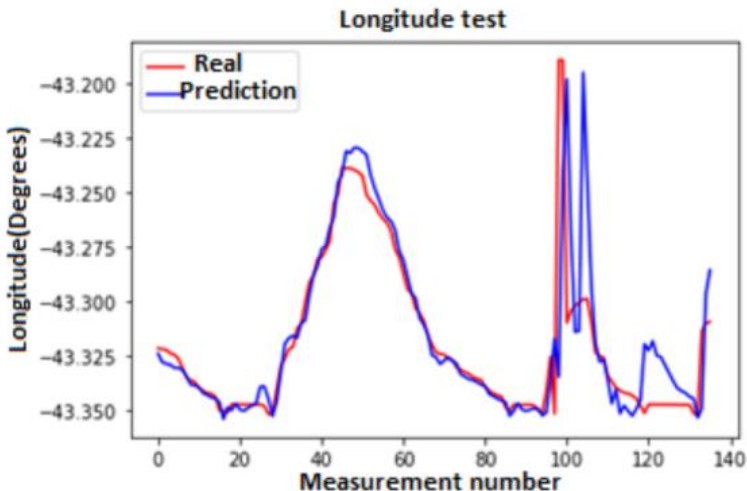

**Figure A2.** Longitude prediction from historical data for a route in Rio de Janeiro.

The second test consisted of predicting the location (latitude and longitude) of a vehicle using the locations generated by the output as inputs, that is, not in a validation environment but with the goal of knowing if the algorithm would be able to make predictions in real-time. The test consisted of using the 186 validation data from the previous test as input values. Of these 186 records, the first 51 data points (from 0 to 50) were chosen. With this data, the algorithm predicts the next location point. Once this has been predicted, and to predict the following points, the previous 51 points are taken, including the one that is just predicted. This means that each time a vehicle's latitude and longitude data were predicted, the data that were previously predicted were used until the 136 corresponding positions of the input dataset were found (which were not initially available).

The algorithm used in these first tests (with a dataset generated in a previous work) has two parts, training and prediction. The training is performed through two ML models; one is trained with the location data and the other one is trained with the time difference in seconds between a certain location point and the previous measurement.

Figures A3 and A4 are the results of the location prediction (latitude and longitude) from the location values predicted by the algorithm. The horizontal axis is in dimensionless units and represents the number of generated measurements while the vertical axis shows the values of latitude and longitude, expressed in decimal degrees. The prediction graph (blue color) has the same behavior as the real value graph (red color) up to the first 51 values. From then on, having used predicted data to obtain future locations, the blue graph "tries" to follow the behavior of the red graph. Although the follow-ups between the real and predicted data are not very close, the graph follows a trend; so, it could be stated that on a transit vehicle route, in which the same locations are repeated periodically, the locations could be satisfactorily predicted. An important aspect to consider is to find, for each route to be managed, an adequate number of data points, both historical and to be predicted.

Once the above was verified, the third test of the sequential LSTM algorithm was performed, which consisted of using two values as input to a LSTM NN model. The first value was the duration between one measurement and the next one obtained from the time record of vehicle C51623. The second input value was the distance between the current position and the previous one. Using these input data, the prediction of the time it would take the vehicle to reach the predicted location (latitude and longitude) was generated as output. Figure A5 is the result of predicting the arrival time at the future location, where the horizontal axis represents the number of measurements made in dimensionless units and the vertical axis shows the time values in seconds. The prediction did not follow the same trend as the real value because the time output values differed by several seconds or minutes. This difference was because some location data points in the dataset had a

following data point with a time difference of several minutes, hours, or even days. This occurs because location capture was not performed in a uniform time interval.

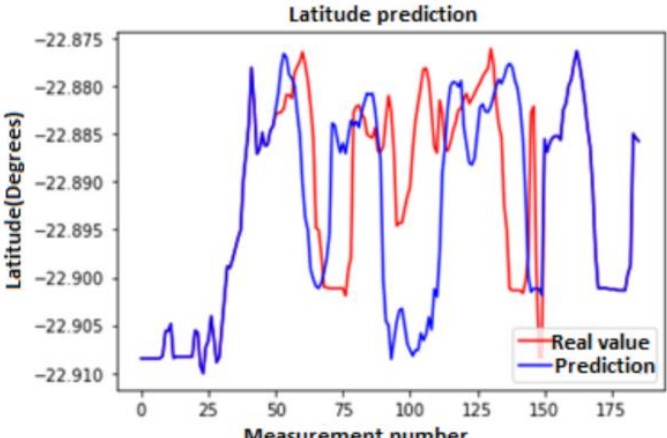

**Figure A3.** Latitude prediction in real-time, based on predicted data for a route in Rio de Janeiro.

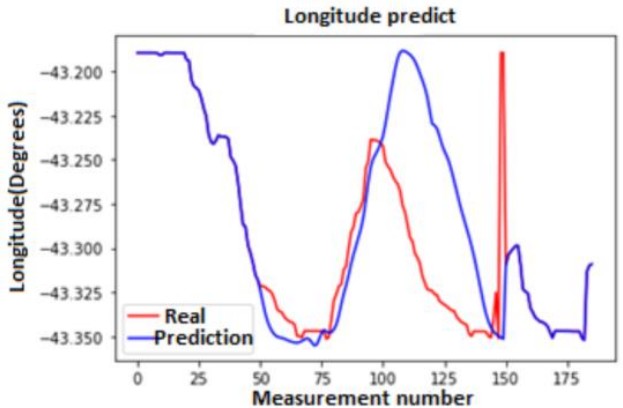

**Figure A4.** Longitude prediction in real-time, based on predicted data for a route in Rio de Janeiro.

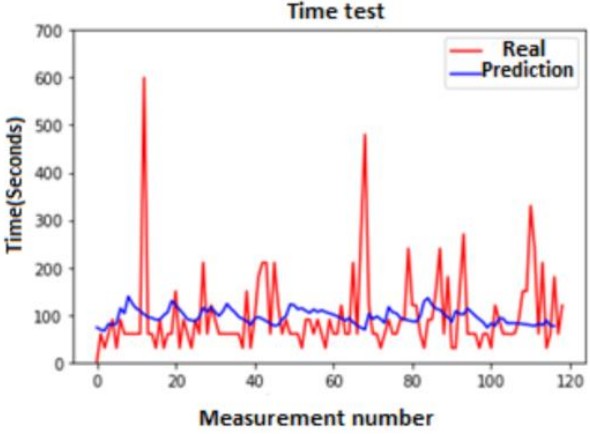

**Figure A5.** Time prediction from historical data for a route in Rio de Janeiro.

*Appendix B.2. Initial Tests Using a Dataset Generated through a Route*

Given the difficulty of not being able to adequately predict the time with the existing dataset [34], a route was traveled to the west of the city of Popayán (Colombia) aboard a vehicle. A Cube Cell HTCC AB02S microcontroller card was located inside the vehicle, which allowed for capturing the measurements from its GPS and IMU sensors to later be

sent through the LoRa communications module to a LoRaWAN gateway, at time intervals from 9 to 13 s, obtaining a more uniform record of the values detected by the sensors. Figure A6 represents the device inside the vehicle. The route is described below:

- Gateway location coordinates: 2.450235, −76.626354;
- Duration: 50 min;
- Traveled distance: 27 km.

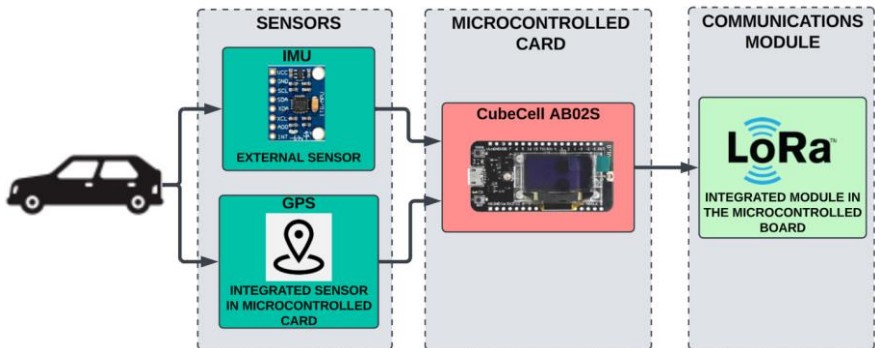

**Figure A6.** Block diagram of the device used to collect the data (initial tests).

The data received by the gateway were stored in a LoRaWAN server. There are several options for this type of server in the market; in this case, The Things Network (TTN [35]) was used. Because the information received using the LoRaWAN was stored for a limited time on the TTN server, another private server was used to store the records permanently in a MYSQL database hosted in the cloud. By tracking the vehicle on the indicated route, 279 records were obtained. The table where the data were stored contains the following columns: id, dev_id, date, hour, latitude, longitude, speed, accx, accy, accz, gyrox, gyroy, and gyroz, where "acc" represents the acceleration of the vehicle at the instant in which the data are sent and "gyro" represents the rate of rotation measured by the gyroscope. Initially, the vehicle speed was used as the input variable; but, during initial tests, it was observed that speed values greater than zero were obtained, even when the device was stationary. Therefore, the speed variable delivered by the GPS was replaced by the acceleration variable delivered by the IMU. Figure A7 represents the process used to receive and store the data.

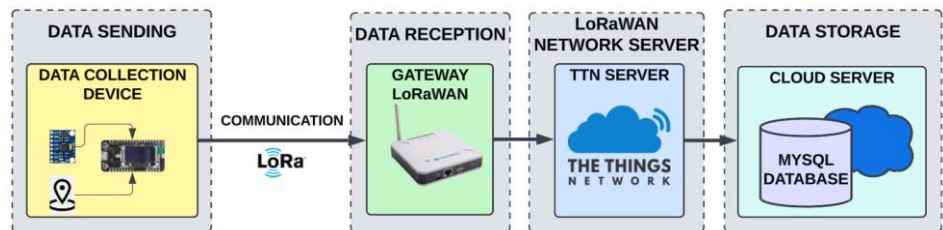

**Figure A7.** Block diagram of data reception and storage (initial tests).

Of the 279 obtained data points, the first 17 were discarded because the value of the latitude and longitude was zero. This is because the GPS module takes some time to connect with the satellites. Thus, 262 valid records were obtained. The data were sent from the device located in the vehicle to the gateway at intervals ranging from 9 to 13 s. The average interval was 11 s and the test duration was 50 min, for which approximately 272 records were expected. Only 262 data points were obtained, most likely due to a loss of the line of sight between the data collection device and the gateway.

For the tests with this dataset obtained through data collection, two prediction scenarios were considered with their respective tests for the execution of the algorithm.

The case in which the information packet sent from the vehicle to the gateway does not arrive, in which case it is not relevant whether a GPS signal is received or not, is identified in Scenario 1, presented in Figure A8. The case when there is no GPS signal available,

but there is LoRa communication between the device and the gateway, is identified in Scenario 2, presented in Figure A9.

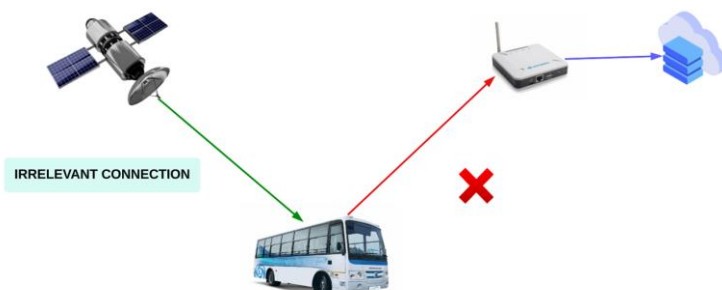

**Figure A8.** Scenario 1. System without LoRa communication, absence of packets in the gateway (it does not matter if there is a GPS connection or not).

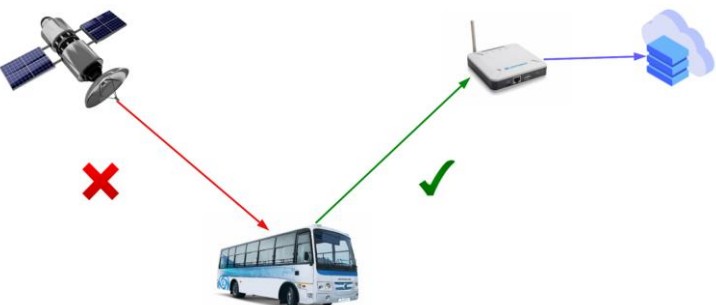

**Figure A9.** Scenario 2. System without a GPS signal and active LoRa communication.

*Appendix B.3. Precision Test of the Algorithm for Scenarios 1 and 2 with a Dataset Generated through a Route*

This test with the new dataset consisted of predicting the location and time based on historical values to measure the accuracy of the used model. Of the 262 obtained data points, the first 210 values were used as training, from which blocks of 30 consecutive data (input vectors) were taken to train the algorithm. The last 82 data points were used for validation. Figures A10 and A11 represent the predicted data against actual values, obtaining values close to each other.

Figures A10 and A11 represent the results of predicting the location (latitude and longitude, respectively) of the vehicle from historical data. In Figures A10 and A11, the first 30 validation values were not plotted because there was no correspondence with the predicted data. For this reason, the graphs go up to measurement number 52.

The mean error values between the predicted position and the actual position for latitude were 0.0003369 and for longitude were 0.0006885; these values in terms of distance, correspond to 85.26 m of mean error.

Each location data point is associated with a timestamp because it is necessary to predict the next location and the time in which the vehicle reaches that location. In this case, for time prediction (Figures A12 and A13), the vertical axis shows how much time the data capture device takes to make a measurement and the horizontal axis shows the number of measurements from the moment the prediction begins. The predicted data are in the range of 9 to 13 s, which corresponds to the time interval in which the location data are sent from the LoRa device to the gateway. To obtain an adequate prediction of the time, the collected values should not be a long time apart. Figure A13 zooms into the concentration of points in Figure A12 to visualize the results in more detail.

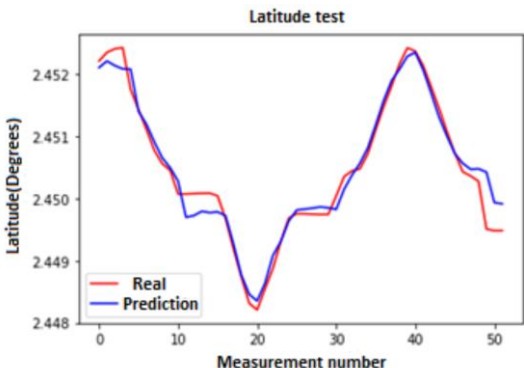

**Figure A10.** Latitude prediction from historical data for a route in Popayán.

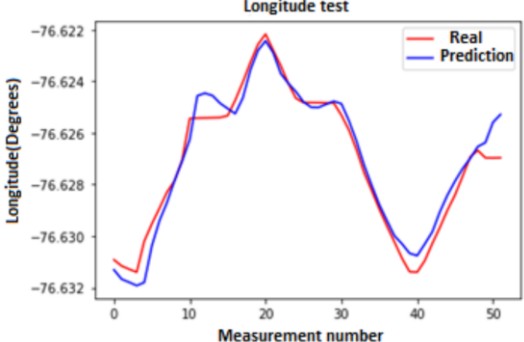

**Figure A11.** Longitude prediction from historical data for a route in Popayán.

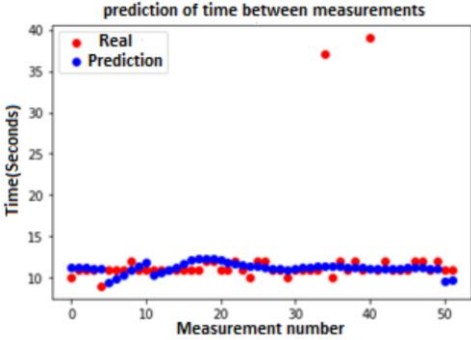

**Figure A12.** Time prediction from historical data for a route in Popayán.

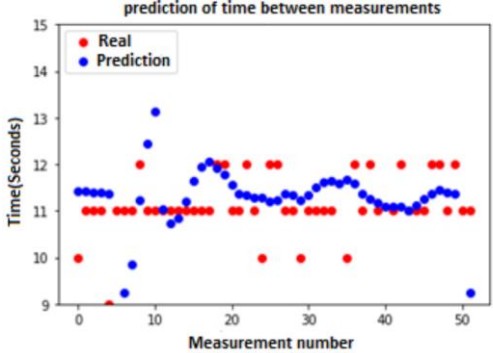

**Figure A13.** Time prediction from historical data for a route in Popayán (enlargement).

The average error value obtained between the predicted times and real times was 6.45 s. This is because there are a couple of time records outside of the 9–13 s interval (Figure A13), indicating that between one measurement and another, packets stopped being sent.

This precision test of the algorithm allowed us to conclude that the mean error values obtained (location and time) between the predicted values and real values are acceptable because, in a real environment, this difference still allows an approximation of where the target vehicle would be located.

Appendix B.3.1. Initial Tests with a Dataset Generated through a Route in Scenario 1

This scenario occurs when there is no communication through LoRa. In this case, information from the GPS or from the IMU is irrelevant because it cannot be transmitted to the gateway, nor stored in a database.

This test with the new dataset predicted the latitude, longitude, and time from the predicted data. The test consisted of using the 82 validation data points from the previous precision test as input values to the LSTM network. Of these 82 records, the first 31 (from 0 to 30) data points were chosen. With these data, the algorithm predicts the next location point; to predict the following points, the previous 30 points are taken, including the one that has just been predicted. This means that each time a vehicle's latitude and longitude data were predicted, the previously predicted data were considered until the remaining positions were found.

The results of predicting the location are observed in Figures A14 and A15. The graphs have the same behavior up to the first 31 measurements; from then on, having used predicted data, the prediction graph (blue color) tries to follow the behavior of the actual value graph (red color). The values between these curves differ by a few decimal places, indicating a difference of meters between the actual and predicted locations (latitude and longitude).

The mean error values between the predicted position and the real position for latitude were 0.0006073 and for longitude were 0.0013994, corresponding to 169.68 m of mean error.

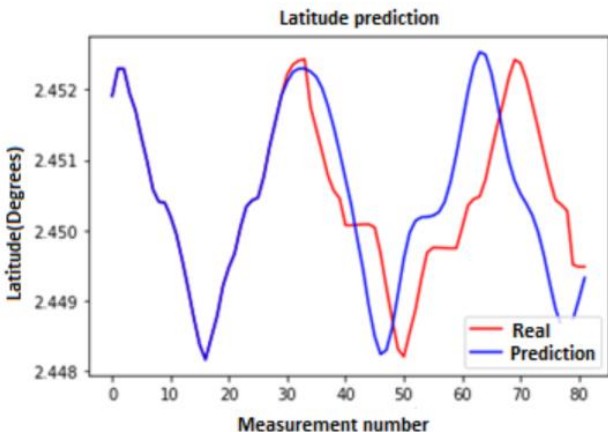

**Figure A14.** Initial latitude prediction from other predicted data for a route in Popayán (in scenario 1).

Figures A16 and A17 show the time prediction. The real value and prediction graphs do not show variations in the first 30 measurements because these first data are used to predict the next position. From then on, the predicted times take values between 11 and 12 s. In addition to concluding that the data points should be close enough in time, it was also possible to conclude that the predicted time tends to be constant within the frequency interval with which the data are sent. Figure A17 zooms into the concentration of the points in Figure A16 to better visualize the results.

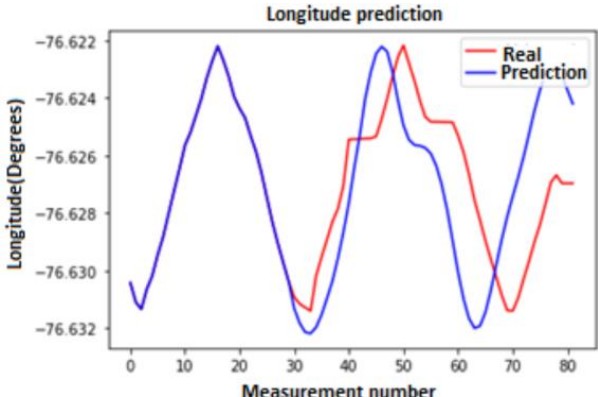

**Figure A15.** Initial longitude prediction from other predicted data for a route in Popayán (in scenario 1).

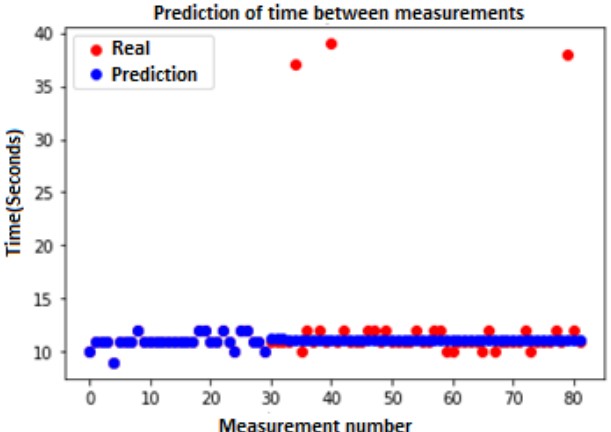

**Figure A16.** Time prediction from other predicted data for a route in Popayán in Scenario 1, initial tests.

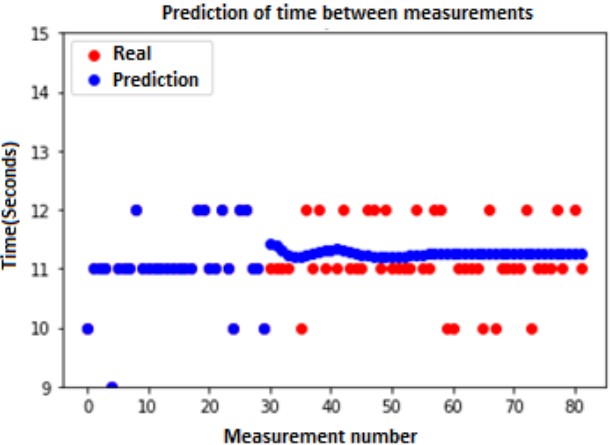

**Figure A17.** Time prediction from other predicted data for a route in Popayán in Scenario 1, initial tests (enlargement).

The average error value obtained between the predicted and real times was 6.48 s.

In conclusion, good results were obtained with variations in distance between the real and predicted data because, in spite of the existence of these variations, the predicted path does not stray from the route taken by the vehicle. At the same time, it was possible to predict the time it would take a vehicle to reach the predicted location, indicating that the algorithm can have good results when making real-time predictions.

In this second scenario, predictions are made when there is a LoRa connection but no GPS communication. In this case, it is not possible to receive location measurements but they could be estimated from IMU data.

The values of latitude, longitude, acceleration, and rotation are used as inputs for this test. Figures A18 and A19 show the trend of values for latitude and longitude; the first 30 data points are the same because they are used to predict the next one, number 31. Then this data point is incorporated into a new entry of 30 data points to find prediction number 32, and so on.

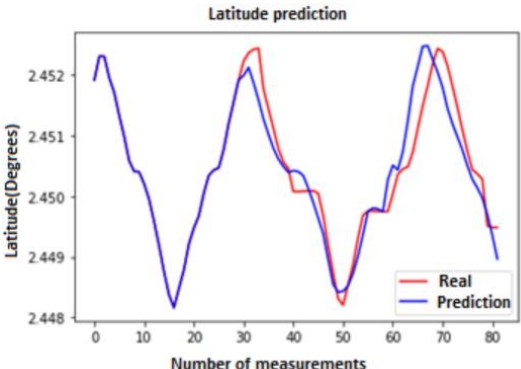

**Figure A18.** Latitude prediction from other predicted data for a route in Popayán without GPS communication (Scenario 2, initial tests).

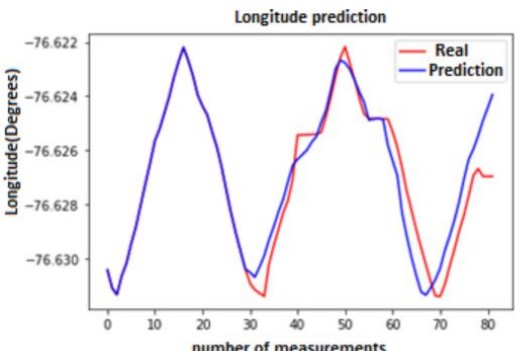

**Figure A19.** Longitude prediction from other predicted data for a route in Popayán without GPS communication (Scenario 2, initial tests).

Calculating the average error between the predicted data and the real data, a value of 0.0004314 is obtained for latitude and a value of 0.0010742 is obtained for longitude; this difference represents 128.75 m.

**Appendix C**

The two mentioned datasets (tables) in Section 3.3.2 are presented in: https://docs.google.com/document/d/1Ovv1PPYBlrmnpW1GxBLItzDQL427krov/edit accessed on 13 September 2023.

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
