# Peer review of "Prototype of a System for Tracking Transit Service Based on IoV, ITS, and Machine Learning"

_wevj, doi:10.3390/wevj14090261_

Round 1

Reviewer 1 Report

This research proposes a transit vehicle tracking system based on IoV in V2R classification. The topic is interesting. The proposed methods have a certain novelty. However, the paper also has room for improvement. Some concerns are listed as follows:

1. The purpose of the paper should be highlighted in Abstract. For example, who can use the method or results proposed in this study

2. Section 2 is suggested to be further improved. The literatures should be further enriched.  

3. In Section 3, it would be better to further highlight your improvement of the method and your innovation in methods.

4. In Section 4 and 5, it would be better to discuss what your findings are different from the past works.

5. It is suggested to further enhance the contributions of the work in Section 6.

Minor editing of English language required.

Reviewer 2 Report

This manuscript needs to be shorter and better organized. It should be shortened and improved.

The introduction section must be improved. It needs to be better organized, and there needs to be narrative order and subject coherence.

The literature gap is not well explained and not satisfactory. More evidence and explanations are required.

The results section must be reorganized. The current structure needs to clarify what type of ML problem is solved (regression or classification); all necessary performance metrics must also be given.

Section 4.3 generated data set must be explained in earlier sections, and also it deserves a better explanation to readers. All statistical properties of the dataset must be given in detail. For example, in line 926, what is the coverage? Is it recall? How was the precision value calculated? (I don't mean the table given in Appendix C)

How LSTM and DBLSTM were trained?

Was k-fold cross-validation applied?

For the ML part of the manuscript, The authors used machine learning features selected only from their systematic literature review; the authors must use several feature selection algorithms in order to detect the best features specific to their problem. It has to be scientifically proven that selected features are the best subset of the dataset.

How can we be sure those algorithms were not overfitted?

LSTM and DBLSTM sections must be better explained, especially the DBLSTM section.

What was the architecture of LSTM and DBLSTM in the author's study? How did they train their LSTMi DBLSTM? How many epochs? What optimization method was used? How many neurons were used?

For Appendix A, what are the hyper-parameters of the best models for each algorithm? Was any hyperparameter selection algorithm applied?

Why were the equations of MEAN, AA, RMSE, MAPE, MAD, ACC-1 BLEU ... etc., metrics selected?

Why are only several metrics applied for several algorithms or methods in Table A? Is there any reason?

The discussion section is not well organized. It should be reorganized and discuss the results and compare theirs' with the other results in the literature.

Round 2

Reviewer 2 Report

Thank you for all the clarification and explanations.

This version of the manuscript was improved.